# Mapping the conformational landscape of a dynamic enzyme by multitemperature and XFEL crystallography

Daniel A Keedy[1†], Lillian R Kenner[1†], Matthew Warkentin[2†], Rahel A Woldeyes[1†], Jesse B Hopkins[2], Michael C Thompson[1], Aaron S Brewster[3], Andrew H Van Benschoten[1], Elizabeth L Baxter[4], Monarin Uervirojnangkoorn[5,6], Scott E McPhillips[4], Jinhu Song[4], Roberto Alonso-Mori[7], James M Holton[3,4,8], William I Weis[5,9,10], Axel T Brunger[5,6,9,10], S Michael Soltis[4], Henrik Lemke[7], Ana Gonzalez[4], Nicholas K Sauter[3], Aina E Cohen[4], Henry van den Bedem[4*], Robert E Thorne[2*], James S Fraser[1*]

[1]Department of Bioengineering and Therapeutic Sciences, University of California, San Francisco, San Francisco, United States; [2]Department of Physics, Cornell University, Ithaca, United States; [3]Physical Biosciences Division, Lawrence Berkeley National Laboratory, Berkeley, United States; [4]Stanford Synchrotron Radiation Lightsource, SLAC National Accelerator Laboratory, Menlo Park, United States; [5]Department of Molecular and Cellular Physiology, Stanford University, Stanford, United States; [6]Howard Hughes Medical Institute, Stanford University, Stanford, United States; [7]Linac Coherent Light Source, SLAC National Accelerator Laboratory, Menlo Park, United States; [8]Department of Biochemistry and Biophysics, University of California, San Francisco, San Francisco, United States; [9]Department of Structural Biology, Stanford University, Stanford, United States; [10]Department of Photon Science, SLAC National Accelerator Laboratory, Menlo Park, United States

*For correspondence: vdbedem@ slac.stanford.edu (HvdB); ret6@ cornell.edu (RET); jfraser@ fraserlab.com (JSF)

†These authors contributed equally to this work

**Abstract** Determining the interconverting conformations of dynamic proteins in atomic detail is a major challenge for structural biology. Conformational heterogeneity in the active site of the dynamic enzyme cyclophilin A (CypA) has been previously linked to its catalytic function, but the extent to which the different conformations of these residues are correlated is unclear. Here we compare the conformational ensembles of CypA by multitemperature synchrotron crystallography and fixed-target X-ray free-electron laser (XFEL) crystallography. The diffraction-before-destruction nature of XFEL experiments provides a radiation-damage-free view of the functionally important alternative conformations of CypA, confirming earlier synchrotron-based results. We monitored the temperature dependences of these alternative conformations with eight synchrotron datasets spanning 100-310 K. Multiconformer models show that many alternative conformations in CypA are populated only at 240 K and above, yet others remain populated or become populated at 180 K and below. These results point to a complex evolution of conformational heterogeneity between 180—240 K that involves both thermal deactivation and solvent-driven arrest of protein motions in the crystal. The lack of a single shared conformational response to temperature within the dynamic active-site network provides evidence for a conformation shuffling model, in which exchange between rotamer states of a large aromatic ring in the middle of the network shifts the conformational ensemble for the other residues in the network. Together, our multitemperature analyses and XFEL data motivate a new generation of temperature- and time-resolved experiments to structurally characterize the dynamic underpinnings of protein function.

**eLife digest** Proteins are the workhorses of the cell. The shape that a protein molecule adopts enables it to carry out its role. However, a protein's shape, or 'conformation', is not static. Instead, a protein can shift between different conformations. This is particularly true for enzymes – the proteins that catalyze chemical reactions. The region of an enzyme where the chemical reaction happens, known as the active site, often has to change its conformation to allow catalysis to proceed. Changes in temperature can also make a protein shift between alternative conformations. Understanding how a protein shifts between conformations gives insight into how it works.

A common method for studying protein conformation is X-ray crystallography. This technique uses a beam of X-rays to figure out where the atoms of the protein are inside a crystal made of millions of copies of that protein. At room temperature or biological temperature, X-rays can rapidly damage the protein. Because of this, most crystal structures are determined at very low temperatures to minimize damage. But cooling to low temperatures changes the conformations that the protein adopts, and usually causes fewer conformations to be present.

Keedy, Kenner, Warkentin, Woldeyes et al. have used X-ray crystallography from a very low temperature (-173°C or 100 K) to above room temperature (up to 27°C or 300 K) to explore the alternative conformations of an enzyme called cyclophilin A. These alternative conformations include those that have previously been linked to this enzyme's activity. Starting at a low temperature, parts of the enzyme were seen to shift from having a single conformation to many conformations above a threshold temperature. Unexpectedly, different parts of the enzyme have different threshold temperatures, suggesting that there isn't a single transition across the whole protein. Instead, it appears the way a protein's conformation changes in response to temperature is more complex than was previously realized. This result suggests that conformations in different parts of a protein are coupled to each other in complex ways.

Keedy, Kenner, Warkentin, Woldeyes et al. then performed X-ray crystallography at room temperature using an X-ray free-electron laser (XFEL). This technique can capture the protein's structure before radiation damage occurs, and confirmed that the alternative conformations observed were not affected by radiation damage.

The combination of X-ray crystallography at multiple temperatures, new analysis methods for identifying and measuring alternative conformations, and XFEL crystallography should help future studies to characterize conformational changes in other proteins.

## Introduction

Current structural biology methods provide only incomplete pictures of how proteins interconvert between distinct conformations (*Motlagh et al., 2014*; *van den Bedem and Fraser, 2015*). Although X-ray crystallography reveals atomic coordinates with relatively high accuracy and precision, the resulting electron density maps contain contributions from multiple alternative conformations reflecting the ensemble average of $10^6$–$10^{15}$ copies of the protein in one crystal (*Rejto and Freer, 1996*; *Smith et al., 1986*; *Woldeyes et al., 2014*). At high resolution, it is often possible to detect and discretely model these alternative conformations (*Burnley et al., 2012*; *Davis et al., 2006*; *Lang et al., 2010*; *van den Bedem et al., 2009*). Structural characterization of alternative conformations by X-ray crystallography can complement NMR (*Baldwin and Kay, 2009*; *Fenwick et al., 2014*) and computational simulations (*Dror et al., 2012*; *Ollikainen et al., 2013*) in defining the structural basis of protein dynamics and ultimately in linking dynamics to function (*Henzler-Wildman and Kern, 2007*).

However, more than 95% of crystal structures are determined at cryogenic temperatures (~100 K) to reduce radiation damage by minimizing diffusion of reactive intermediates and chemical-damage-induced structural relaxations (*Garman, 2010*; *Holton, 2009*; *Warkentin et al., 2013*). Unfortunately, cryocooling can modify main chain and side chain conformational distributions throughout the protein, including at active sites and distal regions important for allosteric function (*Fraser et al.,*

2011; *Halle, 2004*; *Keedy, et al., 2014*). Recent studies have instead used room temperature data collection to reveal a multitude of previously 'hidden' alternative conformations that are not evident at cryogenic temperatures, many of which have important ramifications for determining molecular mechanisms (*Deis et al., 2014*; *Fraser et al., 2009*; *Fukuda and Inoue, 2015*; *van den Bedem et al., 2013*).

Between these temperature extremes, protein conformational heterogeneity changes in complex ways. Previous studies using a wide variety of biophysical probes including NMR, X-ray crystallography, and neutron scattering have revealed a change in the character of conformational heterogeneity and/or protein dynamics around 180–220 K (*Doster, 2010*; *Frauenfelder et al., 2009*; *Lewandowski et al., 2015*; *Ringe and Petsko, 2003*) however, the molecular origins of this 'glass' or 'dynamical' transition remain incompletely understood. Classic work has examined the temperature dependence of protein conformational heterogeneity across individual X-ray structures determined at temperatures from ~80 to 320 K (*Frauenfelder et al., 1979*, *1987*; *Tilton et al., 1992*). These studies used atomic B-factors as a proxy for conformational heterogeneity and identified a global inflection point around 180–220 K. This inflection point was interpreted in terms of a transition driven by dynamical arrest of the coupled hydration layer-protein system (*Doster et al., 1989*; *Frauenfelder et al., 1979*, *1987*; *Tilton et al., 1992*). By contrast, solution NMR studies of picosecond–nanosecond (ps–ns) timescale methyl side chain order parameters showed heterogeneous changes in motional amplitudes at temperatures between 288 and 346 K. Thermal deactivation of these motions was suggested to predict a transition near 200 K without invoking solvent arrest (*Lee and Wand, 2001*). Recent solid-state NMR (ssNMR) experiments suggest that protein motions are coupled to solvent, and that three transitions at ~195, 220, and 250 K mark the onset of distinct classes of motions as temperature increases (*Lewandowski et al., 2015*). Unfortunately, these studies used either globally averaged data (as with ssNMR or neutron scattering) or imprecise atomic-level models of conformational heterogeneity (as with B-factors in X-ray crystallography or NMR order parameters), thus preventing an all-atom understanding of the complex temperature response of protein crystals.

New crystallographic and computational techniques now enable a more detailed investigation of the temperature dependence of protein conformational heterogeneity at the atomic level. First, the program Ringer (*Lang et al., 2014*, *2010*) evaluates low-level electron density traditionally considered noise to uncover statistically significant 'hidden' alternative conformations, which may become populated or depopulated as a function of temperature. Second, multiconformer models with explicit alternative conformations of both backbone and side chain atoms, as created by manual building or methods such as the program qFit (*Keedy et al., 2015*, *van den Bedem et al., 2009*), can account for non-harmonic motions across separate energy wells (encoded by discrete alternative conformations with distinct occupancies and coordinates) and harmonic motions within energy wells (encoded by B-factors). Third, crystallographic order parameters ($S^2$) weigh these harmonic and non-harmonic contributions in a single metric that quantifies the disorder of each residue in a multiconformer model, allowing direct comparison with NMR-determined order parameters (*Fenwick et al., 2014*). Finally, methodological advances based on the physics of ice formation have enabled variable-temperature crystallographic data collection at temperatures between 300 and 100 K with modest or no use of potentially conformation-perturbing cryoprotectants (*Warkentin et al., 2012*; *Warkentin and Thorne, 2009*). Together, these methods overcome many of the limitations of previous X-ray-based approaches and will contribute to an integrated view of how protein conformational heterogeneity and dynamics evolve with temperature.

The human proline isomerase cyclophilin A (CypA) is an excellent model system for deploying these tools to study the structural basis of functional conformational dynamics and, in particular, to use temperature to understand the extent of correlated motions during an enzyme's catalytic cycle. Previous NMR relaxation data for CypA (*Eisenmesser et al., 2005*, *2002*) indicated a single common exchange process, both in the apo state and during catalysis, for a network of dynamic residues extending from the core to the active site. Room temperature crystallography later suggested the precise alternative conformations that collectively interconvert during catalysis (*Fraser et al., 2009*). However, subsequent NMR relaxation experiments of mutants designed to perturb the dynamics suggested that multiple exchange processes occur within this network (*Schlegel et al., 2009*). Here, we analyze multitemperature synchrotron experiments to examine the temperature-dependent conformational heterogeneity of CypA. Additionally, we report X-ray-free electron laser

(XFEL) data, which are free of conventional radiation damage (*Kern et al., 2014*; *Spence et al., 2012*), to validate previous connections between alternative conformations determined by synchrotron crystallography and NMR experiments performed in solution (*Eisenmesser et al., 2005*; *Fraser et al., 2009*). Our analysis shows that the temperature dependence of alternative protein conformations is heterogeneous and that the character of this heterogeneity bridges previous models for protein dynamical transitions. Our results also suggest new ways to use variable temperature with both synchrotron and XFEL crystallography to probe the dynamic underpinnings of protein function.

## Results

### Multitemperature X-ray datasets reveal modulated conformational ensembles of CypA

To probe the conformational landscape of CypA, we collected eight high-resolution (1.34 –1.58 Å) synchrotron crystallographic datasets across a wide range of temperatures from 100 to 310 K (*Table 1*) with no added cryoprotectants. For each dataset, we initially refined single-conformer models. Although the single-conformer models are very similar to each other, the accompanying electron density maps reveal differences throughout the protein. In the active-site network, the mFo-DFc difference electron density maps are relatively featureless below 200 K, suggesting that a single conformation is a valid fit below this temperature. By contrast, positive and negative mFo-DFc peaks become gradually more prevalent as temperature increases above 200 K, suggesting that multiple conformations are increasingly required to explain the data as temperature increases (*Figure 2—figure supplement 1*).

We monitored the shift from single-conformation to multiple conformations both visually (*Figure 1A,B*) and using the automated electron density scanning program Ringer (*Figure 1C,D*). Briefly, Ringer identifies alternative conformations at low levels of electron density by evaluating the density value for the γ atom at each possible position about the χ1 dihedral angle, given a fixed main chain conformation (*Lang et al., 2014*, *2010*). We focused on two residues, Ser99 and Leu98, which are key markers of the conformational exchange by NMR (*Eisenmesser et al., 2002*, *2005*) and were implicated in our previous room-temperature X-ray and mutagenesis experiments (*Fraser et al., 2009*). For both Ser99 (*Figure 1A*) and Leu98 (*Figure 1B*), a dominant peak is evident at all temperatures. The reduced height of this peak as temperature increases is accompanied by the increase in a secondary peak corresponding to the electron density of the minor conformation. To quantify this trend, we computed correlation coefficients between the electron density versus dihedral angle curves for each residue (*Figure 1C,D*). Pairs of curves for similar temperatures have higher correlations than those for different temperatures. In particular, pairs of curves for temperatures that span the low-temperature (100–180 K) and high-temperature (240–310 K) regimes are more poorly correlated than are curves from the same temperature regime. The dynamical transitions observed in previous studies (*Doster, 2010*; *Lee and Wand, 2001*; *Lewandowski et al., 2015*; *Ringe and Petsko, 2003*; *Schiro et al., 2015*) generally occur between these two temperature regimes.

To ground this conformational redistribution in all-atom detail, we built a multiconformer model with qFit (*Keedy et al., 2015*; *van den Bedem et al., 2009)* for each multitemperature dataset. We then finalized the model by manually editing alternative conformations and refining to convergence, resulting in models that were improved relative to the single-conformer models (*Table 2*, *Video 1*).

At 180 K and below, the active-site network is best modeled as a single state, with electron density corresponding to ordered water molecules clearly evident adjacent to Phe113 (*Figure 2*, top row). At 240 K and above, by contrast, multiple conformations provide a better explanation of the data. Interestingly, some partial-occupancy water molecules are still present and likely co-occur with the major conformations (*Figure 2*, middle and bottom rows). Met61 appears to populate additional conformations above 180 K, although it is difficult to precisely define changes in its conformational ensemble as temperature increases. This residue bridges Phe113 and the catalytic residue Arg55 via steric contacts between alternative conformations in both directions, emphasizing the importance of modeling multiple conformations in all-atom detail for understanding inter-residue coupling.

**Table 1.** Crystallographic statistics for multitemperature synchrotron datasets collected on a single crystal per dataset. Statistics for the highest resolution shell are shown in parentheses.

| | 100 K | 150 K | 180 K | 240 K | 260 K | 280 K | 300 K | 310 K |
|---|---|---|---|---|---|---|---|---|
| PDB ID | 4YUG | 4YUH | 4YUI | 4YUJ | 4YUK | 4YUL | 4YUM | 4YUN |
| Wavelength (Å) | 0.9767 | 0.9767 | 0.9767 | 0.9767 | 0.9767 | 0.9767 | 0.9767 | 0.9767 |
| Resolution range (Å) | 33.58–1.48 (1.53–1.48) | 16.95–1.34 (1.39–1.34) | 16.12–1.38 (1.43–1.38) | 34.05–1.42 (1.47–1.42) | 33.98–1.48 (1.53–1.48) | 25.23–1.42 (1.47–1.42) | 22.67–1.5 (1.55–1.50) | 22.66–1.58 (1.64–1.58) |
| Space group | $P2_12_12_1$ | $P2_12_12_1$ | $P2_12_12_1$ | $P2_12_12_1$ | $P2_12_12_1$ | $P2_12_12_1$ | $P2_12_12_1$ | $P2_12_12_1$ |
| Unit cell (a, b, c) | 42.24, 51.91, 88.06 | 42.45, 51.82, 88.01 | 42.42, 51.96, 88.21 | 43.04, 53.22, 88.63 | 43.09, 52.79, 88.81 | 43.00, 52.61, 89.12 | 43.01, 52.61, 89.32 | 42.85, 52.58, 89.41 |
| Total reflections | 160,129 (15,842) | 160,780 (7,437) | 154,202 (11,295) | 152,578 (13,600) | 134,699 (13,381) | 168,932 (15,019) | 144,734 (14,433) | 125,225 (12,326) |
| Unique reflections | 32,657 (3,240) | 42,288 (3,471) | 39,548 (3,820) | 38,881 (3,710) | 34,411 (3,391) | 38,763 (3,794) | 32,999 (3,254) | 28,291 (2,760) |
| Multiplicity | 4.9 (4.9) | 3.8 (2.1) | 3.9 (3.0) | 3.9 (3.7) | 3.9 (3.9) | 4.4 (4.0) | 4.4 (4.4) | 4.4 (4.5) |
| Completeness (%) | 99 (100) | 95 (80) | 97 (95) | 99 (96) | 100 (100) | 100 (100) | 99 (100) | 100 (100) |
| Mean I/sigma (I) | 14.07 (1.57) | 25.95 (3.24) | 16.47 (1.64) | 12.86 (1.66) | 10.09 (1.46) | 15.51 (1.52) | 16.90 (1.63) | 13.26 (1.45) |
| Wilson B-factor (Å$^2$) | 16.07 | 13.12 | 16.95 | 15.55 | 16.06 | 17.62 | 19.75 | 21.44 |
| R-merge (%) | 6.8 (99.4) | 3.0 (29.4) | 4.2 (71.8) | 6.2 (99.2) | 8.1 (104.3) | 4.9 (100.0) | 4.7 (101.7) | 6.7 (127.3) |
| R-measurement (%) | 7.6 (111.0) | 3.4 (36.9) | 4.8 (85.6) | 7.2 (116.8) | 9.4 (120.8) | 5.6 (115.3) | 5.4 (115.9) | 7.6 (144.5) |
| CC1/2 | 1.00 (0.62) | 1.00 (0.90) | 1.00 (0.60) | 1.00 (0.50) | 1.00 (0.52) | 1.00 (0.52) | 1.00 (0.59) | 1.00 (0.56) |
| CC* | 1.00 (0.88) | 1.00 (0.97) | 1.00 (0.87) | 1.00 (0.82) | 1.00 (0.83) | 1.00 (0.83) | 1.00 (0.86) | 1.00 (0.85) |
| Refinement resolution range (Å) | 33.085–1.48 (1.558–1.48) | 19.117–1.34 (1.394–1.34) | 16.995–1.38 (1.435–1.38) | 34.055–1.42 (1.477–1.42) | 33.98–1.48 (1.547–1.48) | 25.23–1.42 (1.477–1.42) | 22.67–1.5 (1.579–1.5) | 25.2221.58 (1.679 –1.58) |
| Reflections used in refinement | 32,627 (4,654) | 42,278 (3,932) | 39,545 (4,265) | 38,879 (4,161) | 34,411 (4,237) | 38,762 (4,256) | 32,999 (4,643) | 28,287 (4,632) |
| Reflections used for R-free | 1,028 (147) | 1,325 (125) | 1,238 (133) | 1,218 (130) | 1,080 (133) | 1,217 (133) | 1,036 (145) | 889 (146) |
| R-work (%) | 13.3 (20.4) | 12.4 (16.4) | 13.3 (25.4) | 12.6 (26.3) | 13.1 (26.0) | 11.1 (22.6) | 10.8 (20.0) | 11.7 (21.8) |
| R-free (%) | 18.3 (26.8) | 15.6 (21.3) | 17.5 (33.0) | 15.6 (30.4) | 16.8 (31.2) | 14.3 (25.5) | 14.4 (24.8) | 15.0 (28.8) |
| Number of non-hydrogen atoms | 2,279 | 2,433 | 1,969 | 1,993 | 2,035 | 2,120 | 2,096 | 2,172 |
| Macromolecule atoms | 1,933 | 2,132 | 1,745 | 1,750 | 1,837 | 1,924 | 1,952 | 2,061 |
| Protein residues | 165 | 164 | 164 | 163 | 163 | 163 | 163 | 163 |
| RMS (bonds) (Å) | 0.009 | 0.008 | 0.008 | 0.009 | 0.009 | 0.008 | 0.009 | 0.009 |
| RMS (angles) (°) | 1.16 | 1.20 | 1.23 | 1.20 | 1.16 | 1.16 | 1.14 | 1.14 |
| Ramachandran favored (%) | 97 | 94 | 97 | 96 | 97 | 96 | 97 | 96 |
| Ramachandran allowed (%) | 3.3 | 5.7 | 2.7 | 4.1 | 3 | 4.2 | 3.3 | 3.9 |
| Ramachandran outliers (%) | 0 | 0 | 0 | 0 | 0 | 0 | 0 | 0 |
| Rotamer outliers (%) | 2.4 | 1.3 | 0.53 | 1.1 | 1.5 | 1.9 | 1.4 | 0.88 |
| Clashscore | 0.57 | 1.08 | 0.00 | 1.24 | 0.27 | 0.78 | 0.52 | 0.00 |
| Average B-factor (Å$^2$) | 21.74 | 17.25 | 21.85 | 20.14 | 20.00 | 21.48 | 24.09 | 25.77 |
| Macromolecule average B-factor (Å$^2$) | 18.48 | 14.67 | 19.99 | 17.95 | 18.17 | 19.61 | 22.82 | 24.94 |
| Solvent average B-factor (Å$^2$) | 39.99 | 35.54 | 36.34 | 35.89 | 37.01 | 39.89 | 41.23 | 41.30 |

PDB: Protein Data Bank. CC: correlation coefficient.

## XFEL data confirm conformational heterogeneity in synchrotron data is not due to radiation damage

Quantifying radiation damage versus exposure dose (*Figure 1—figure supplement 1*) and limiting exposure dose per dataset ensured that the conformational heterogeneity observed in multitemperature synchrotron datasets was not dominated by radiation damage. However, XFELs can generate data that are entirely free from conventional radiation damage by diffraction-before-destruction data collection (*Kern et al., 2014*; *Spence et al., 2012*). To compare the distribution of alternative conformations between synchrotron and XFEL data, we collected two ambient-temperature datasets: a 1.75 Å resolution radiation-damage-free dataset using serial femtosecond rotation crystallography (*Table 3*) (*Hirata et al., 2014*; *Schlichting, 2015*; *Suga et al., 2015*) and an additional 1.2 Å resolution synchrotron dataset (*Table 4*). For the XFEL experiment, we collected 1,239 individual diffraction images, translating to unique unexposed regions of 71 crystals between each shot (*Video 2*), and processed them using cctbx.xfel (*Hattne et al., 2014*) with post-refinement in PRIME (*Uervirojnangkoorn et al., 2015*). Automated molecular replacement yielded interpretable electron density maps that allowed us to refine a single-conformer structural model with reasonable

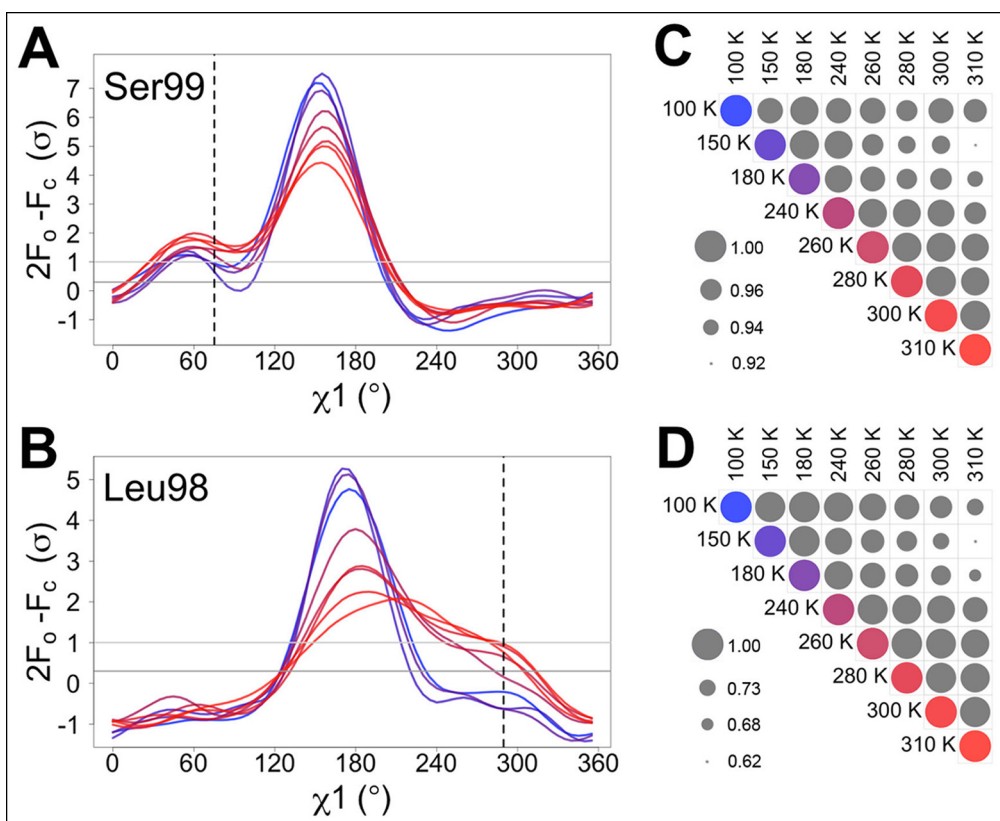

**Figure 1.** Automated electron density sampling reveals increased conformational redistribution. Ringer curves of 2mFo-DFc electron density versus χ1 dihedral angle for (**A**) Ser99 and (**B**) Leu98 show large peaks for modeled major conformations and smaller peaks for additional minor conformations (dashed vertical lines). These secondary peaks become more evident as temperature increases (color gradient from blue to purple to red). A backrub motion was used for Ser99. For (**C**) Ser99 and (**D**) Leu98, a Pearson correlation coefficient was calculated between each pair of Ringer curves from the corresponding panel in (**A**) or (**B**). Circles in diagonal elements are colored as in (**A**) or (**B**); circles in off-diagonal elements are all gray but scaled by pairwise correlation coefficient (see legend). Pairs of curves from similar temperatures are generally more correlated to each other (larger circles) than are pairs of curves from more different temperatures (smaller circles).

The following figure supplement is available for figure 1:

**Figure supplement 1.** Radiation damage is minimal across data collection temperatures.

**Table 2.** Improvements in validation statistics from finalizing raw qFit models. Statistics calculated with *phenix.molprobity*.

| | | RT synchrotron | XFEL | 100 K | 150 K | 180 K | 240 K | 260 K | 280 K | 300 K | 310 K |
|---|---|---|---|---|---|---|---|---|---|---|---|
| $R_{free}$ (%) | Raw qFit | 16.7 | 25.2 | 19.0 | 16.9 | 18.5 | 17.5 | 17.9 | 15.7 | 16.3 | 16.1 |
| | Final | 14.6 | 24.9 | 18.3 | 15.6 | 17.5 | 15.6 | 16.8 | 14.3 | 14.4 | 15.0 |
| | Δ | −2.1 | −0.3 | −0.7 | −1.3 | −1.0 | −1.9 | −1.1 | −1.4 | −1.9 | −1.1 |
| MolProbity score | Raw qFit | 1.47 | 1.80 | 1.79 | 1.31 | 1.21 | 1.18 | 1.45 | 1.28 | 0.95 | 1.19 |
| | Final | 1.08 | 1.39 | 1.19 | 1.29 | 0.63 | 1.14 | 0.91 | 1.25 | 0.99 | 0.76 |
| | Δ | −0.39 | −0.41 | −0.80 | −0.02 | −0.58 | −0.04 | −0.54 | −0.03 | 0.04 | −0.43 |

RT: Room temperature; XFEL: X-ray-free electron laser.

quality statistics. Electron density sampling analysis using Ringer and multiconformer refinement using qFit were performed as for the multitemperature synchrotron data.

In agreement with our previous room-temperature studies (*Fraser et al., 2009*), the XFEL and synchrotron mFo-DFc difference maps reveal evidence for the rate-limiting alternative conformations extending from the active site into the core of the protein (*Figure 3A,B*). For example, the backrub-coupled (*Davis et al., 2006*) rotamer jump of Phe113 is apparent from a large positive mFo-DFc peak in both maps. Alternative conformations for core residue Ser99 are also evident from mFo-DFc peaks (*Figure 3A,B*) and Ringer electron density sampling curves (*Figure 3E*). We did not conclusively observe a secondary peak in the electron density sampling curve corresponding to a discrete alternative conformation of Leu98 (*Figure 3F*), but that is likely due to the lower resolution of the XFEL dataset. Multiconformer models for both datasets (*Figure 3C,D*) again feature alternative conformations across the active-site network and are strongly supported by 2mFo-DFc electron density. These results provide an important positive control on the observation of conformational heterogeneity in our synchrotron studies by establishing that electron density corresponding to the alternative conformations of CypA is not an artifact of radiation damage. The ability of XFEL crystallography to reveal native and functionally important alternative conformations at high resolution may be especially useful for other systems that are presently intractable for room- or variable-temperature synchrotron crystallography due to the small size of available crystals.

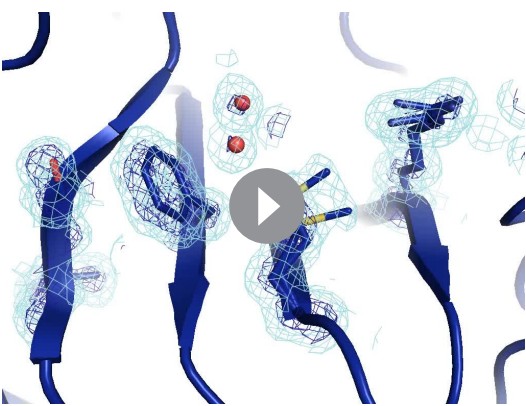

**Video 1.** Animated interpolation between electron density maps in temperature trajectory. For each pair of adjacent temperatures (e.g. 100 and 150 K), the temperature regime between them was bisected and an average 2mFo-DFc electron density map was calculated in reciprocal space using CCP4 utilities, until temperature points were spaced by <1 K. A new multiconformer model is shown when the animation reaches the corresponding temperature.

## Some regions feature conformational heterogeneity only at low temperatures

Although more conformational heterogeneity is expected with our higher temperature synchrotron datasets, and is evident in the active site of CypA, cooling can also stabilize new conformations (*Halle, 2004*). For example, the loop containing residues 79–83 (*Figure 4*, *Video 3*) exhibits conformational heterogeneity only at cryogenic temperatures. This region is well fit by a single conformation at 240 K and above, but a secondary loop conformation is necessary to explain the electron density at 100, 150, and 180 K. Additionally, the loop is clearly single-state in the highest resolution (1.2 Å) dataset (*Figure 4—figure supplement 1*), demonstrating that the slightly lower resolution of the elevated-temperature datasets does not obscure a secondary conformation.

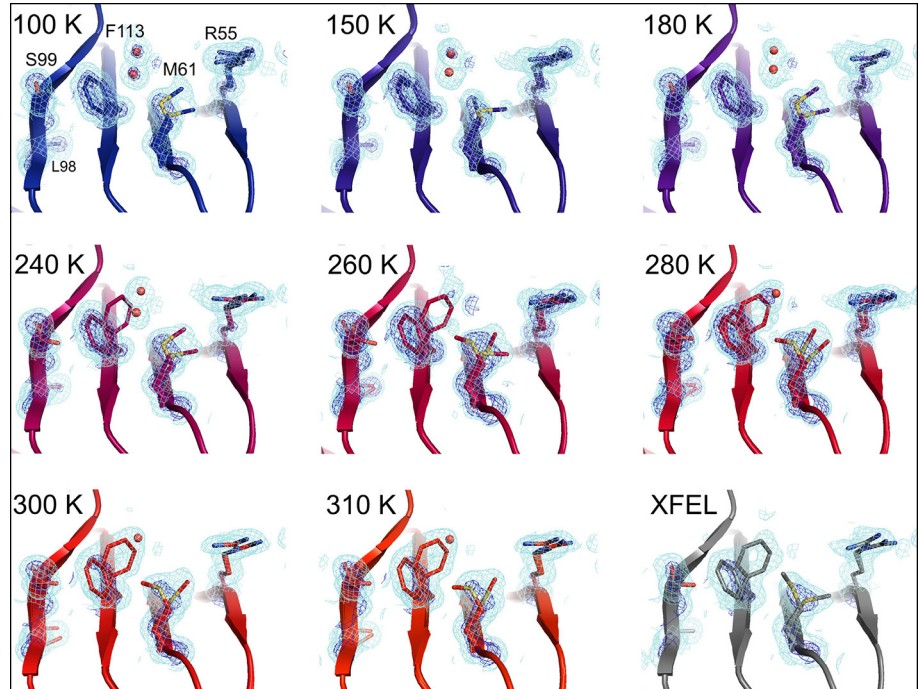

**Figure 2.** Multiconformer modeling across temperatures captures increasing conformational heterogeneity. Residues extending from the core to the active site of cyclophilin A (CypA) adopt a single conformer at low temperatures, but gradually transition to increasing occupancy of secondary conformations as temperature increases. These conformations are well supported by 2mFo-DFc electron density contoured at 0.6 σ (cyan mesh) and 3.0 σ (dark blue mesh). This is corroborated by the room-temperature X-ray free-electron laser (XFEL) model (gray), which is free from conventional radiation damage and features the same secondary conformations. Water molecules (red spheres) are more fully ordered at low temperatures, but become only partially occupied at higher temperatures because they are mutually exclusive with the secondary Phe113 conformation.

The following figure supplement is available for figure 2:

**Figure supplement 1.** Single-conformer models cannot explain the crystallographic data at higher temperatures..

In the primary conformation, the 79–83 loop is not involved in any main chain–main chain hydrogen bonds to the rest of CypA, suggesting that the barrier to forming the secondary conformation does not involve breakage of cooperative secondary-structure-like interactions. The observation of a secondary state for residues 79–83 at 100–180 K, but not at 240–310 K, suggests that it is enthalpically stabilized at lower temperatures (*Halle, 2004*; *Keedy et al., 2014*). Consistent with this mechanism, the secondary conformation of the 79–83 loop is accompanied by an ordered, partial-occupancy water molecule (*Figure 4*, top row). This water molecule, which is clearly distinct from the carbonyl oxygen of the primary conformation of Glu81, wedges between the loop and the rest of the protein. The surprising appearance of specific solvent-linked protein conformational heterogeneity exclusively below 240 K emphasizes the complex and heterogeneous changes in protein–solvent energetics that can occur at cryogenic temperatures.

## Quantifying temperature-dependent changes in conformational heterogeneity

Despite counter examples such as the 79–83 loop, most residues in CypA, especially in the active site, exhibit increases in discrete conformational heterogeneity above 180 K. To quantify these changes in regions implicated by NMR relaxation experiments, we measured the 2mFo-DFc electron density in the volumes occupied by the alternative conformations of Ser99 and Phe113. By contrast, B-factors, which can model the harmonic motions near any single conformation, are poor proxies for the non-harmonic change between discretely separated conformations. To quantify the change in

**Table 3.** Crystallographic statistics for room-temperature XFEL dataset collected across 71 crystals. Statistics for the highest resolution shell are shown in parentheses.

| | XFEL |
|---|---|
| PDB ID | 4YUP |
| Resolution range (Å) | 43.98 –1.75 (1.81 –1.75) |
| Space group | P2$_1$2$_1$2$_1$ |
| Unit cell (a, b, c) | 42.42, 51.82, 87.96 |
| Unique reflections | 19,942 (1894) |
| Completeness (%) | 99 (96) |
| Wilson B-factor (Å$^2$) | 21.12 |
| Refinement resolution range (Å) | 43.98 –1.75 (1.93 –1.75) |
| Reflections used in refinement | 19,936 (4,811) |
| Reflections used for R-free | 625 (151) |
| R-work (%) | 20.0 (34.3) |
| R-free (%) | 24.9 (36.1) |
| Number of non-hydrogen atoms | 1,762 |
| Macromolecular atoms | 1,559 |
| Protein residues | 164 |
| RMS (bonds) (Å) | 0.017 |
| RMS (angles) (°) | 1.44 |
| Ramachandran favored (%) | 96 |
| Ramachandran allowed (%) | 3.6 |
| Ramachandran outliers (%) | 0 |
| Rotamer outliers (%) | 1.8 |
| Clashscore | 1.92 |
| Average B-factor (Å$^2$) | 29.03 |
| Macromolecule average B-factor (Å$^2$) | 26.52 |
| Solvent average B-factor (Å$^2$) | 48.25 |
| Number of TLS groups | 3 |

PDB: Protein Data Bank; TLS: translation libration screw; XFEL: X-ray-free electron laser.

minor state occupancy as a function of temperature, we summed the electron density in the volume that is occupied exclusively by the minor conformation and avoided any voxels that overlap with the van der Waals radii of atoms of the major conformation (*Figure 5A*). The resulting curves of minor-state electron density versus temperature have a shallow slope at 180 K and below, but a much steeper slope at 240 K and above (*Figure 5B,C*). Additionally, the electron density for the XFEL data is consistent with the data collection temperature (273 K) and the overall trends.

However, most residues that populate alternative conformations do not have such easily characterized and separable regions of electron density. To quantify how conformational heterogeneity throughout CypA varies as a function of temperature, we used B-factor-dependent crystallographic order parameters (S$^2$) (*Fenwick et al., 2014*). These order parameters include both harmonic contributions, which reflect conformational heterogeneity near one conformation (encoded by B-factors), and non-harmonic contributions, which reflect conformational heterogeneity between multiple discretely separated conformations (encoded by occupancies and displacements in coordinates). Importantly, these order parameters account for both conformational heterogeneity within energy wells, whether it is modeled by B-factors or by subtly different alternative conformations, as well as discretely separated alternative conformations that occupy distinct rotamers. Similar to the

**Table 4.** Crystallographic statistics for room-temperature synchrotron dataset collected on a single crystal. Statistics for the highest resolution shell are shown in parentheses.

|  | 1.2 Å Synchrotron |
| --- | --- |
| PDB ID | 4YUO |
| Wavelength (Å) | 0.9795 |
| Resolution range (Å) | 44.60 –1.20 (1.24 –1.20) |
| Space group | $P2_12_12_1$ |
| Unit cell (a, b, c) | 42.9, 52.43, 89.11 |
| Total reflections | 307,722 (18,999) |
| Unique reflections | 58,118 (5,122) |
| Multiplicity | 5.3 (3.7) |
| Completeness (%) | 91 (82) |
| Mean I/sigma (I) | 10.99 (5.93) |
| Wilson B-factor (Å$^2$) | 15.22 |
| R-merge (%) | 11.2 (20.4) |
| R-measurement (%) | 12.2 (23.4) |
| CC1/2 | 0.99 (0.96) |
| CC* | 1.00 (0.99) |
| Refinement resolution range (Å) | 45.19 –1.20 (1.23 –1.20) |
| Reflections used in refinement | 58,108 (3,657) |
| Reflections used for R-free | 2,000 (126) |
| R-work (%) | 12.7 (31.3) |
| R-free (%) | 14.6 (33.5) |
| Number of non-hydrogen atoms | 2327 |
| Macromolecular atoms | 2143 |
| Protein residues | 163 |
| RMS (bonds) (Å) | 0.009 |
| RMS (angles) (°) | 1.16 |
| Ramachandran favored (%) | 96 |
| Ramachandran allowed (%) | 4.1 |
| Ramachandran outliers (%) | 0 |
| Rotamer outliers (%) | 0.84 |
| Clashscore | 0.98 |
| Average B-factor (Å$^2$) | 19.62 |
| Macromolecule average B-factor (Å$^2$) | 18.40 |
| Solvent average B-factor (Å$^2$) | 33.86 |

PDB: Protein Data Bank.

2mFo-DFc electron density integration results for Phe113, we observed a large change in χ1 bond order parameters at 240 K and above (*Figure 6A*).

## Mapping the transitions in CypA conformational heterogeneity

Next, we applied the order parameter analysis to all side chain χ1 angles in CypA. Although conformational heterogeneity generally increases with temperature throughout the enzyme, we observed a diverse set of conformational responses (*Figure 6—figure supplement 1*). The trends for the majority of residues suggested a transition somewhere between our data points at 180 and 240 K, below which the change in conformational heterogeneity with temperature is reduced. To quantify this

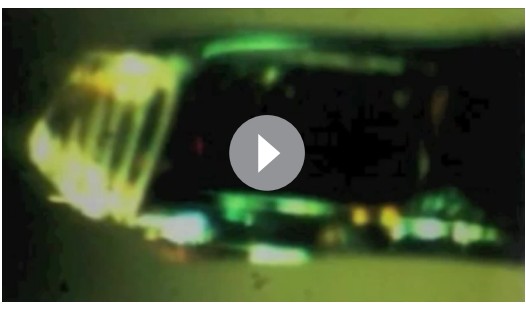

**Video 2.** Fixed-target X-ray-free electron laser (XFEL) data collection from cyclophilin A (CypA) crystals at the LCLS-XPP end station. Screen capture image of the Blu-Ice GUI showing a video display of a CypA crystal. After each shot, a new damage line appears and the crystal is translated.

trend, we performed separate fits for the low-temperature ($\leq$180 K) and high-temperature ($\geq$240 K) data points for all residues. The slopes of conformational heterogeneity ($1 - S^2$) versus temperature were significantly different ($p=1 \times 10^{-62}$, paired T-test) on either side of this transition range: the average slope for the low-temperature fit lines ($2.5 \times 10^{-4}$ K$^{-1}$) was an order of magnitude smaller than for the high-temperature fit lines ($2.6 \times 10^{-3}$ K$^{-1}$). This is consistent with the idea that heterogeneity is much less dependent on temperature below the 180–240 K 'transition' range.

However, some residues behaved differently from the rest of the protein. Val2 retains its conformational heterogeneity at all temperatures, which is expected based on its weakly constrained position at the N terminus. Many of the remaining outlier residues (Glu15, Glu81, Pro105, Ala117, Glu120, Lys125, Met142, Ser147, Lys151) appear to be involved in a spatially contiguous set of crystal contacts across symmetry mates in the context of the crystal lattice (*Figure 6—figure supplement 2*). This cluster includes Glu81, which adopts alternative backbone conformations

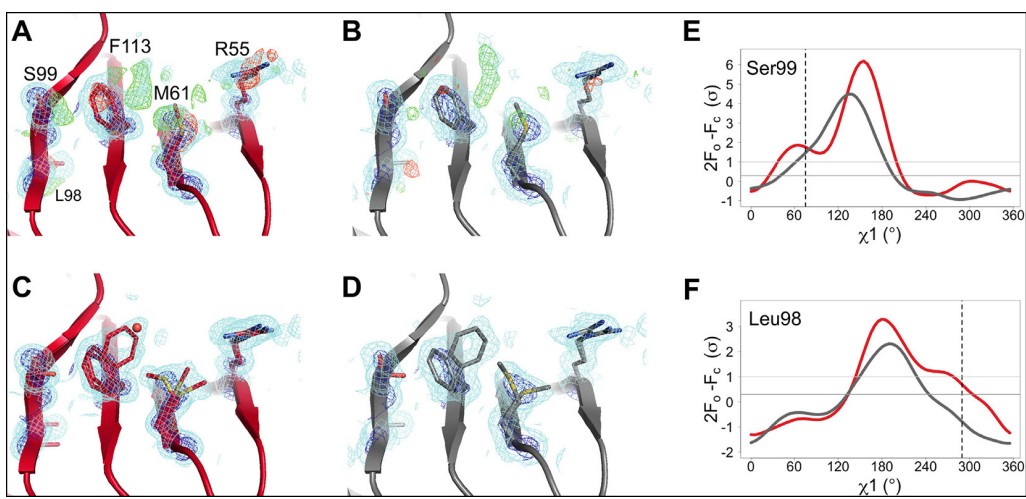

**Figure 3.** The active-site conformational ensemble of cyclophilin A (CypA) determined without radiation damage at room temperature. (**A**) Electron density maps for room-temperature synchrotron (red) and (**B**) X-ray-free electron laser (XFEL) (silver) single-conformer models reveal conformational heterogeneity extending from the protein core (Leu98 and Ser99) to the active site (Arg55) of CypA. The primary conformation is well supported by 2mFo-DFc electron density contoured at 0.6 σ (cyan mesh) and 3.0 σ (dark blue mesh). mFo-DFc difference electron density contoured at 3.0 σ (green mesh) and − 3.0 σ (red mesh) suggests unmodeled alternative conformations. (**C, D**) Finalized multiconformer models explicitly model these alternative conformations, which are well-supported by 2mFo-DFc electron density. (**E, F**) Ringer electron density sampling for the single-conformer models shows peaks representing alternative conformations for (**E**) Ser99 and (**F**) Leu98. The primary conformations of both residues are obvious as peaks for both models, but the minor conformations (dashed vertical line; as modeled in 3k0n) are also evident, with 2mFo-DFc values well above the 0.3σ (darker gray horizontal line) threshold, except for the Leu98 in the XFEL model (due to the lower resolution). A backrub motion of −10° positions the backbone properly for Ringer to best detect the minor conformation for Ser99, but not for Leu98.

The following figure supplement is available for figure 3:

**Figure supplement 1.** The 1.2 Å room-temperature CypA synchrotron data show no signs of radiation damage.

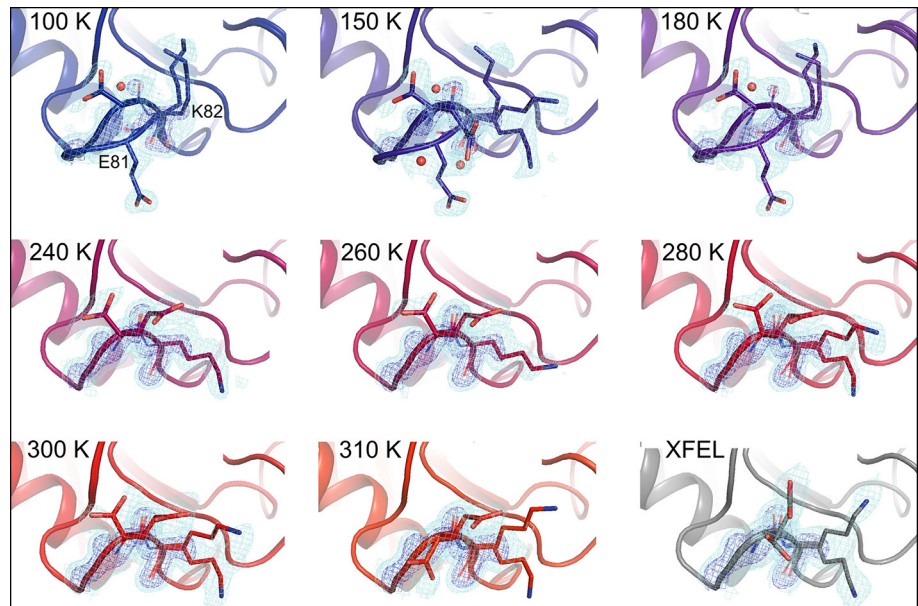

**Figure 4.** Alternative loop conformations can appear at lower temperatures. The surface loop containing residues 79–83 adopts alternative conformations at low temperatures (top row) but not at high temperatures (bottom two rows). The secondary loop conformation is separated from the body of the protein by an ordered water molecule (red sphere); the van der Waals interactions between the loop and the water may reflect an enthalpic stabilization that is more dominant at low temperatures. The electron density peak to the right of the water corresponds to the backbone carbonyl oxygen of Glu81. 2mFo-DFc electron density contoured at 0.6σ (cyan mesh) and 2.0σ (dark blue mesh). XFEL: X-ray-free electron laser.
The following figure supplement is available for figure 4:

**Figure supplement 1.** Alternative loop conformations are not present in the highest-resolution (1.2 Å) dataset.

only at low temperatures (*Figure 4*). The variability of these residues can likely be explained by distinct sets of conformations across crystal contacts that are differentially, but somewhat stochastically, favored during the cooling process (*Alcorn and Juers, 2010*).

Our data suggest that CypA does not undergo a single global transition from having strongly temperature-dependent changes in side chain conformational heterogeneity to relatively temperature-independent behavior. An 'intersection' or 'transition' temperature for each bond angle can be estimated from the intersection of the low-temperature and high-temperature fit lines of $1 - S^2$ versus temperature. The distribution of these intersection temperatures is broad and asymmetrical, with an elongated tail from the peak near 250 K toward 200 K (*Figure 6B*). Furthermore, the distribution of intersection temperatures is more complex for order parameters reporting on the terminal heavy-atom bond of the side chain than for $\chi 1$ (*Figure 6—figure supplement 3,4*). This increase likely occurs because side chain end orientations are subject to more degrees of freedom and therefore temperature changes may redistribute them in a greater variety of ways.

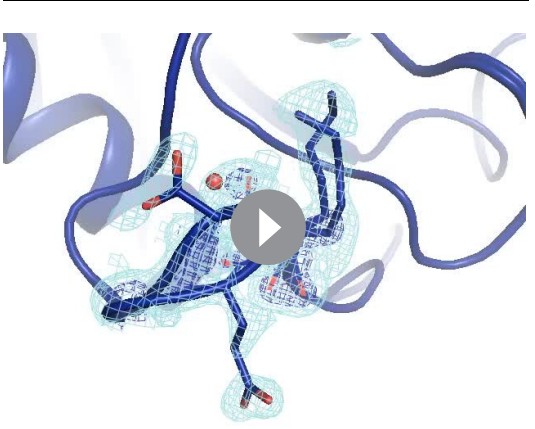

**Video 3.** Animated rotation around the 100 K (blue) and 310 K (red) models and electron density maps from *Figure 4*.

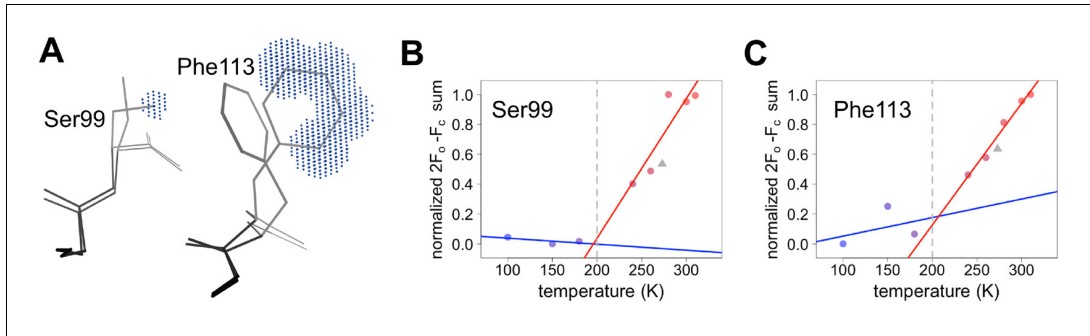

**Figure 5.** Quantifying temperature titration of conformational heterogeneity in multiconformer models. (**A**) 2mFo-DFc electron density was summed over the volume occupied by the minor conformation but not the major conformation (blue grid points) for Ser99 and Phe113. (**B,C**) Minor-state 2mFo-DFc electron density increases with temperature. Electron density sums were normalized for each residue. Multitemperature points from synchrotron data are shown in colors corresponding to temperature. The X-ray-free electron laser point is shown as a gray triangle. Best-fit lines are shown for 180 K and below (blue) versus 240 K and above (red).

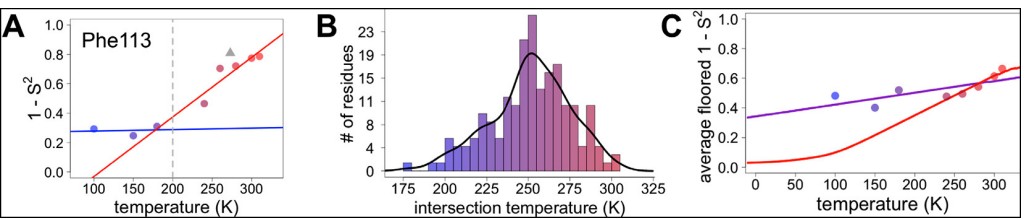

**Figure 6.** Diversity in temperature dependences of side chain disorder across cyclophilin A (CypA) does not predict the observed average arrest of disorder. (**A**) The complement of B-factor-influenced side chain order parameter for the bond most closely associated with the χ1 dihedral angle for Phe113. Lines reflect least-squares fits to synchrotron models at 180 K and below (blue) versus 240 K and above (red). Multitemperature synchrotron points in colors; X-ray-free electron laser (XFEL) point (not included in fits) as gray triangle. (**B**) Distribution of the intersection temperature between the <200 and >200 K lines fitted with kernel density function. The peak is near 250 K, although there is a tail toward lower temperatures. Intersection temperatures were <170 K for four residues and >330 K for five residues. (**C**) Predicted and observed values for the complement of side chain order parameter, averaged over all residues in CypA. The predicted values were obtained by extrapolating each residue's fit line for 240 K and above (red curve) or for the full 100–300 K (purple curve), flooring the result to 0, then averaging across all residues in CypA. Observed values, similarly floored and averaged, are shown as points.

The following figure supplements are available for figure 6:

**Figure supplement 1.** Heterogeneous response of side chain disorder to temperature.

**Figure supplement 2.** Residues with persistent disorder across temperatures are interconnected in the crystal lattice.

**Figure supplement 3.** Diversity in temperature dependences of side chain-end disorder across CypA does not predict the observed average arrest of disorder.

**Figure supplement 4.** Heterogeneous response of side chain-end disorder to temperature.

**Figure supplement 5.** Different residues extrapolate to different maximal-order temperatures.

**Figure supplement 6.** Both harmonic and non-harmonic flexibilities contribute to changes in order parameters with temperature.

**Figure supplement 7.** Globally averaged disorder exhibits an apparent transition near 250 K.

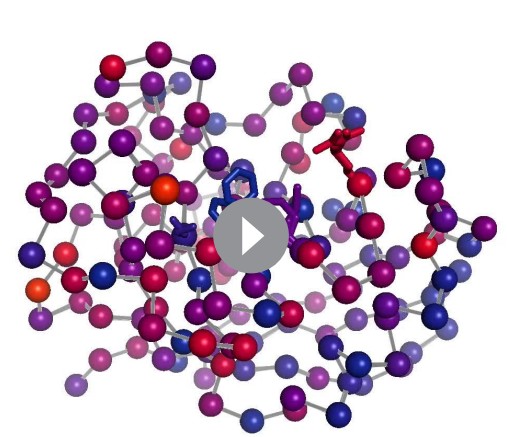

**Figure 7.** The temperature dependence of side chain disorder is non-homogenously spatially distributed in CypA. Intersection temperatures from (A) χ1 order parameters as in *Figure 6B* or (B) side chain terminus order parameters as in *Figure 6—figure supplement 3* B are mapped to the 1.2 Å room-temperature synchrotron model. Each residue is marked with a sphere colored based on its apparent transition temperature, from low (blue) to high (red). The active-site network is subdivided: Ser99 and Phe113 (left of boxed region) both transition at a low temperature regardless of order parameter bond vector, but Met61 and Arg55 transition at higher, different temperatures.

## Distinguishing between models of protein heterogeneity as a function of temperature

Our data provide insight into models for the origin of the temperature dependence of protein conformational heterogeneity and into proposed dynamical transitions. In one model, deactivation of different internal protein motions at high temperatures (near 300 K) is sufficient to predict a dynamical transition near 200 K (*Lee and Wand, 2001*). In a second model, solvent-coupled arrest of protein motions produces a transition in a similar temperature range (*Ringe and Petsko, 2003*). To distinguish between these two models, we analyzed the average side chain disorder across all residues in CypA, focusing on the bond most closely associated with the χ1 dihedral angle, at each of the eight temperatures we studied (*Figure 6C*). These averaged disorder values drop as temperature is decreased from 310 K, then flatten out somewhere between 240 and 180 K, with some scatter due to the variability in the cryocooling process (data points in *Figure 6C* and *Figure 6—figure supplement 7*).

Next, we used two different linear fits to extrapolate $1 - S^2$ across all temperatures for each residue, floored the result at maximum order ($1 - S^2 = 0$), and then averaged across all residues to obtain predictions for the residue-averaged disorder versus temperature. In the first fit, the linear function was fit to data for each residue at all temperatures. Consistent with the necessity of using separate fit lines for the low-temperature and high-temperature data (*Figure 5* and *Figure 6—figure supplement 1*), the resulting prediction gives a poor account of the averaged experimental data and does not indicate a transition (purple line in *Figure 6C*). In the second fit, only the high-temperature (240 K and above) data for each residue were fit. The resulting prediction is more consistent with the averaged high-temperature experimental data and does indicate a transition (red line in *Figure 6C*). The flattening of this predicted curve at low temperatures occurs as more individual residues achieve maximal predicted order ($S^2 \rightarrow 1$) (*Figure 6—figure supplement 5*). This latter prediction, which is extrapolated from high-temperature crystallographic data, is reminiscent of predictions based on NMR relaxation experiments conducted at

**Video 4.** 360° rotation around *Figure 7B*.

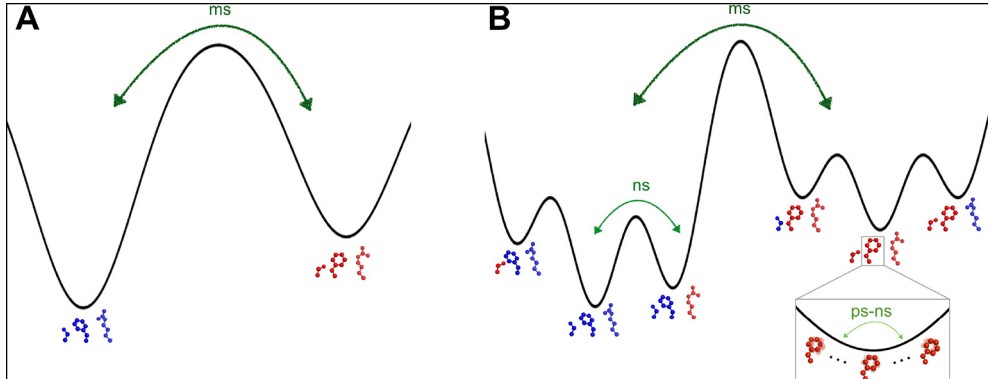

**Figure 8.** The dynamic active-site network of cyclophilin A (CypA) has a complex energy landscape. (**A**) The previous simple model in which Ser99, Phe113, and Arg55 (Met61 omitted for clarity) interconvert from one macrostate (blue) to the other (red) completely collectively. NMR data suggest this process occurs on a millisecond timescale. (**B**) A more nuanced model in which network microstates are populated differently depending on the network macrostate, defined by the Phe113 rotameric state. In the left macrostate, Ser99 rotamer changes are disfavored because of steric overlaps with Phe113, but Arg55 rotamer changes are accommodated; the reverse is true (perhaps to a lesser extent) in the right macrostate. Within each microstate, rapid thermal motions occur (bottom right), and may alleviate some minor steric overlaps. Timescales are estimates consistent with NMR observables for CypA and other systems. ms: millisecond; ns: nanosecond; ps: picosecond.

288–346 K (*Lee and Wand, 2001*). Our observations are consistent with the idea that thermal depopulation of protein alternative conformations is sufficient to predict the existence of an average inflection without invoking a transition of the solvent. However, the low temperature of the predicted inflection (~100 K), as well as the large separation in low-temperature disorder between our experimental data (data points in *Figure 6C*) and predictions from high temperature (red line in *Figure 6C*), suggest that thermal depopulation of protein alternative conformations cannot by itself account for the observed ~200 K transition. This large separation also indicates that data collected at high temperatures (>260 K) cannot be reliably extrapolated to predict conformational heterogeneity at low temperatures. It also follows that data from low temperatures (≤180 K) cannot be simply extrapolated to predict the features of the energy landscape that may be important above ~200 K.

Many effects may contribute to the discrepancy between the observed data and the behavior projected from the high-temperature fits. To gain additional insight into this discrepancy, we decomposed the order parameters into their B-factor versus discrete-conformers components and examined their temperature dependences (*Figure 6—figure supplement 6*). Roughly 67% of residues (e.g. Phe8) remain within one rotamer well across all temperatures. Approximately 13% of residues (e.g. Ser99 and Phe113) populate clearly separable multiple rotameric states at high temperatures, and then show complete depopulation of minority states on cooling so that only a single rotameric state remains at 180 K and below. However, 6% of residues (e.g. Thr5) continue to populate multiple rotameric states at or below 180 K. An additional 6% of residues (e.g. Lys91) populate new rotameric states only at or below 180 K. These results help explain the excess residual disorder in our experimental structures below 240 K compared to projections based on high-temperature fits. Although slopes of crystallographic B-factors with temperature remain nearly flat below 240 K (*Figure 6—figure supplement 6B*), we expect that true harmonic thermal disorder does subtly decrease from 180 to 100 K; these thermal effects could be more detectable if even higher resolution data were collected at even lower temperatures, perhaps by using liquid helium as a cryogen to cool to ~15 K or below (*Chinte et al., 2007*).

## Imperfect coupling between active- site residues in CypA

While the results above show a variety of thermal and non-thermal conformational responses, it remains unclear whether these responses involve coupled conformational shifts of multiple residues. In particular, the network of alternative side chain conformations spreading from the core

of the protein (Ser99) into the active site (Arg55) across multiple β-strands exhibits qualitatively similar behavior of increasing occupancy above 240 K. In previous work (*Fraser et al., 2009*), the collective presence of these alternative conformations at room temperature, but not at cryogenic temperatures, and the close contacts between these residues had suggested a concordance with the single exchange process fit by NMR relaxation dispersion for this dynamic active-site network. However, using our new multitemperature data, this network appears subdivided based on the apparent intersection or transition temperatures of the constituent residues, with Ser99 and Phe113 behaving most similarly to each other (*Figure 7*, *Video 4*).

## Discussion

Here, we have mapped the conformational landscape of the dynamic enzyme CypA by analyzing multiconformer models from multitemperature crystallography. Unlike previous temperature-dependent analyses of X-ray crystallography, here we consider both harmonic disorder (B-factors) and non-harmonic displacements (alternative conformations and occupancies), characterized using crystallographic bond order parameters. We have four primary findings:

1. The occupancy of alternative side chain conformations present at 310 K generally decreases with decreasing temperature.
2. However, some residues adopt alternative conformations only at 180 K and below.
3. The temperature response of residues is highly heterogeneous: the 'intersection temperature' below which side chain conformational heterogeneity remains nearly constant has a broad distribution across the molecule, with a peak near 250 K.
4. Residues composing a dynamic active-site network that is important for function do not have a consistent intersection temperature.

Our results provide new insight into the relationship between energy landscapes, the glass transition, and protein function. Glasses and other disordered systems have complex energy landscapes, in part due to the large number of microenvironments and the extensive frustration that disorder generates in the intermolecular interactions. Proteins at biological temperatures are 'glassy' in this sense—they have complex energy landscapes, due to their large size, heterogeneous amino acid composition, and many degrees of conformational freedom (*Frauenfelder et al., 1991*). The extensive heterogeneity in the temperature response of individual residues in CypA that we observe here provides additional direct evidence for this underlying energetic heterogeneity.

In addition to the inherent 'glassiness' of proteins at biological temperatures, dynamical transitions, some of which have been called 'glass transitions', have been reported at lower temperatures, including 180, 200, 220, 240, and 250 K, based on Mössbauer spectroscopy, X-ray crystallography, liquid and solid-state NMR, neutron scattering, and other techniques (*Lewandowski et al., 2015*; *Schiro et al., 2015*). These transitions typically manifest as a change in slope of some measurement in the vicinity of the suggested transition temperature. However, many of these measurements are sensitive to motions only within some timescale window (often ps-ns), monitor only a subset of amino acid types, and/or spatially average over all residues in the protein. By contrast, multitemperature crystallography with multiconformer models has many advantages by providing a time-independent and fully site-resolved measurement of ensemble-averaged atomic displacements, including both harmonic and discrete conformational heterogeneity, within the crystal.

This combined methodology lets us examine dynamical and glass transitions in protein crystals from a new perspective. Glass transitions are by definition non-equilibrium phenomena that arise when the kinetics of relaxation toward equilibrium slow so dramatically that equilibrium cannot be reached on experimental timescales. One signature of a true glass transition in proteins would be if occupancies of minority alternative conformations were arrested at non-zero values below some temperature. Indeed, here we see no appreciable temperature evolution of individual conformer occupancies or B-factors at 180 K and below, and the average disorder at these temperatures is far in excess of what is predicted from high-temperature extrapolations. These observations are consistent with the falling out of equilibrium expected in a glass transition, but not with a transition driven by the thermal freeze-out of alternative side chain conformations (*Lee and Wand, 2001*). Moreover, the persistence of multiple rotameric states at low temperatures is consistent with solvent arrest that impedes further changes in side chain disorder. Although the details of protein–solvent interactions

may differ in crystals versus in solution, local variability at different protein–solvent interface microenvironments in the crystal (*Teeter et al., 2001*) likely contributes to the heterogeneity of temperature responses that we observe. The critical importance of site resolution is evident in the results of *Figure 6*. Averaging over side chain disorder in all residues yields an apparent transition near 250 K (*Figure 6* and *Figure 6—figure supplement 7*). Perhaps coincidentally, this same temperature has been associated with a transition for protein side chains in site-averaged solid-state NMR measurements (*Lewandowski et al., 2015*). However, the 'transition' in our residue-averaged result obscures the highly heterogeneous temperature dependence of the individual side chains. We find no evidence of tight cooperativity or of a collective global response near 250 K that would be expected in the case of a 'true' dynamical transition. Instead, our data are consistent with local, non-cooperative freeze-out of conformational states defined by the energy landscape over a broad temperature range.

The heterogeneous response of side chain order parameters to temperature is driven largely by the changes to the populations of alternative conformations, which 'flat-line' at different temperatures across CypA. This diversity of 'flat-lining' temperatures is present even within the dynamic active-site network, even though the constituent residues have similar occupancies for their major and minor states at high temperatures (*Figure 7*). This result contrasts with previous NMR and X-ray experiments that hypothesized correlated motions of this network as rate-limiting for the catalytic cycle (*Eisenmesser et al., 2005*; *Fraser et al., 2009*) (*Figure 8A*).

To bridge these views, we propose that the active-site network adopts two substates, which are primarily distinguished by Phe113 rotamer interconversion. Each of these substates adopts a differently weighted ensemble of conformations for other residues (*Figure 8B*). In this model, Met61 and Arg55 can switch rotamers more easily than Ser99 when Phe113 is in its χ1 gauche (p) rotamer pointed toward Ser99, whereas Ser99 can switch rotamers more easily than Met61 and Arg55 when Phe113 is in its χ1 gauche (m) rotamer pointed toward Met61. Additionally, thermal 'breathing' motions within rotameric wells may relieve minor steric overlaps within some of these macro- and microstates (*Figure 8B* , bottom right). This model is consistent with Phe113 having the lowest 'flat-lining' temperature of the network (*Figure 7*), and makes sense sterically because of the large size of the aromatic ring. These hypothesized motions are consistent with the timescales and temperature dependencies of motion assigned by solid-state NMR studies of crystalline protein GB1 (*Lewandowski et al., 2015*). Furthermore, the model helps explain the difficulty of fitting NMR relaxation data for perturbed versions of the active-site network as a single collective exchange process (*Schlegel et al., 2009*). The aromatic ring of Phe113 could play a dominant role in determining the chemical shift changes of the surrounding residues. Each of these residues could also populate multiple rotamers in the excited state measured by NMR. Our hierarchical perspective evokes the 'population shuffling' model of (*Smith et al., 2015*), in which a protein macrostate (in CypA, defined by the Phe113 rotamer) also determines the different relative populations of rotamers for a subset of other residues (in CypA, the other residues in the active-site network). In this model, the interconversion between macrostates, and not the collective motion of all residues between distinct rotamers, is correlated with the rate-limiting step of the CypA catalytic cycle.

Diversity in the temperature dependences of alternative conformations as we see here is inevitable given the limitations of the amino acid alphabet, yet its spatial pattern within a protein may provide insight into selective pressures. Evolutionary optimization must ensure that functionally important alternative conformations are robustly populated and interconvert appreciably at physiological conditions. However, the energy landscapes of individual residues are coupled to varying extents, such that some subsets of residues must be collectively optimized to preserve some, but not perfect, collectivity in functional motions. For proteins with large sequence alignments, evolutionary covariation has been used to predict 'sectors' of functionally cooperative residues, which are often dispersed in primary sequence but strikingly contiguous in tertiary structure (*Halabi et al., 2009*). By contrast, temperature-dependent crystallography has the potential to unveil couplings in atomic detail by identifying sets of residues whose conformational ensembles respond concertedly to temperature change. Based on our results with CypA, we expect this coupling to be weak, but measurable. Serial femtosecond XFEL crystallography combined with ultra-fast temperature jumps could enable a temporal view of these coupled conformational changes. Novel static and time-resolved multitemperature crystallographic approaches will provide powerful tools for resolving concerted motions to explore how proteins function and evolve.

# Materials and methods

## Protein expression, purification, and crystallization

Wild-type CypA was produced and crystallized as previously reported (*Fraser et al., 2009*). Briefly, crystals were grown by mixing equal volumes of well solution (100 mM (4-(2-hydroxyethyl)-1-pipera-zineethanesulfonic acid) HEPES pH 7.5, 23% PEG 3350, 5 mM Tris (2-carboxymethyl) phosphine [TCEP]) and protein (60 mg mL $^{-1}$ in 20 mM HEPES pH 7.5, 100 mM NaCl, 0.5 mM TCEP) in the hanging-drop format.

## Crystallographic data collection

For the multitemperature synchrotron datasets at 100, 150, 180, 240, 260, 280, 300, and 310 K, we collected data at the Cornell High Energy Synchrotron Source (CHESS) at beamline A1 with a 100 μm collimator using a wavelength of 0.9767 Å. Crystals were looped by hand, stripped of excess mother liquor (100 mM HEPES pH 7.5, 23% PEG 3350, 5 mM TCEP) using NVH oil (*Warkentin and Thorne, 2009*), and placed directly into the nitrogen-gas cryostream pre-set to the desired temperature at the beamline. Water inside protein crystals is nanoconfined so that ice nucleation is dramatically suppressed, but water outside crystallizes readily and rapidly. Careful removal of all excess solvent from the crystal surface is essential to obtaining ice-free diffraction between 260 K and 180 K without using large cryoprotectant concentrations.

For the XFEL experiment, we collected multiple diffraction images per crystal using a 10-μm X-ray beam with each irradiation point separated by at least 25–40 μm to avoid collateral radiation damage. A total of 1,239 still diffraction images were collected from 71 CypA crystals over the course of two experiments using a goniometer setup and a Rayonix MX325HE detector at LCLS-XPP (*Cohen et al., 2014*) (*Video 2*). All data were collected at ambient temperature (approximately 273 K). To prevent dehydration, crystals were coated with paratone oil immediately after looping and mounted on the goniometer at the XPP endstation of LCLS using the SAM sample exchange robot (*Cohen et al., 2002*).

For the new 1.2 Å room-temperature synchrotron dataset, paratone oil was applied to cover a 2 μL hanging drop containing a single large crystal of CypA. The crystal was harvested through the paratone and excess mother liquor was removed using a fine paper wick. Attenuated data were collected at SSRL beamline 11-1 at 273 K controlled by the cryojet on the PILATUS 6M PAD detector.

## Crystallographic data processing

The synchrotron datasets were indexed, integrated, and scaled using XDS and XSCALE, and intensities were subsequently converted to structure factor amplitudes using XDSCONV. All datasets were from single crystals. Data reduction statistics for the highest resolution room-temperature dataset and the multitemperature datasets can be found in *Tables 1,4* respectively.

The XFEL data were processed using *cctbx.xfel* (*Hattne et al., 2014*). Of the 1,239 images collected, 772 were indexed and their intensities were integrated. Post-refinement, as implemented by *PRIME* (**p**ost-**r**ef**i**nement and **me**rging, version date: November 11, 20:22:51 2014) (*Uervirojnangkoorn et al., 2015*), was used to correct the intensity measurements and merge the data. We optimized over the uc_tolerance, n-postref_cycle, sigma_min, partiality_min, and gamma_e values to obtain the final structure factor amplitudes. Data reduction statistics for the XFEL data are provided in *Table 3* .

To promote consistency between models derived from different datasets, R$_{free}$ flags were generated using *PHENIX* for the highest resolution 'reference' (1.2 Å, 273 K) dataset first and were subsequently copied to all other multitemperature and XFEL datasets for the automated molecular replacement and refinement pipeline.

## Model building

For each dataset, we calculated initial phases by performing molecular replacement with *phenix. auto_mr* using PDB ID 2cpl as a search model. We next refined XYZs and ADPs of the initial model with *phenix.refine* for 4 macrocycles with XYZ and ADP weight optimization turned on; identified translation libration screw (TLS) groups with *phenix.find_tls_groups*; and refined optimized XYZs,

ADPs, and TLS parameters for six more macrocycles. These single-conformer models and associated electron density maps were used as input for two subsequent steps.

First, the single-conformer models were analyzed with Ringer (*Lang et al., 2010*) via *mmtbx. ringer* using default settings. A coupled side chain–backbone 'backrub' motion (*Davis et al., 2006*) of −10° for Ser99 (see *Figure 5A*) was necessary to match the Cα and Cβ positions of the minor conformation as modeled in PDB ID 3k0n; using this modified backbone indeed yielded maximal minor-conformation Ringer peaks for our multitemperature datasets. No backrub motion was necessary for Leu98 due to the different type of backbone displacement (*Fraser et al., 2009*). Correlation coefficients between pairs of Ringer electron density versus dihedral angle curves were calculated using the *cor* function in R (*Team, 2014*).

Second, the single-conformer models were used as input to *qFit* (*Keedy et al., 2015*; *van den Bedem et al., 2009*). Subsequent to the automated model building, we manually deleted ill-fitting waters and edited alternative protein side chain conformations based on fit to the electron density in *Coot* (*Emsley et al., 2010*) and refinement with *phenix.refine*. For example, at 240 K, *qFit* automatically modeled Phe113 as single-state, but significant mFo-DFc peaks remained, so we decided on a two-state model. Met61 was particularly difficult to model accurately across temperatures due to convolved issues of χ3 non-rotamericity for Met in general (*Butterfoss et al., 2005*), the relatively high electron count for sulfur, and likely temperature-modulated Met-specific radiation damage. For these reasons, visual inspection of the maps and manual building is currently essential for alternative backbone conformations with moderate displacements, as observed in residues 79–83 (*Figure 4*). We are currently developing new methods to automatically detect and model such backbone excursions in multiscale multiconformer models. These efforts improved $R_{free}$ and MolProbity scores across datasets (*Table 2*). Because of the lower resolution, the XFEL model was refined with three TLS groups and with optimization of X-ray versus geometry and ADP weights.

## Model and electron density analysis

For minor-state electron density sums, 2mFo-DFc (Fc filled) map values were summed across a grid of points defined by superimposing each model onto PDB ID 3k0n using all Cα atoms, summing the 2mFo-DFc value at each point with 0.25 Å of a target minor-state heavy atom (Oγ for Ser99; Cδ1, Cε1, Cε2, or Cζ for Phe113), and normalizing to unity across datasets for each residue being analyzed. This procedure allowed a strictly common reference set of map evaluation points. Results were very similar when using unfilled maps (data not shown).

We calculated B-factor-influenced order parameters ($S^2$) as previously reported (*Fenwick et al., 2014*) except that we monitored one of two different types of bond vector. For the χ1 order parameter, we used Cβ-Xβ (where X = C or O) for most amino acids, Cα-Cβ for Ala, and Cα-Hα for Gly. For the side chain-end order parameter, we used the heavy-atom to heavy-atom bond vector for each amino acid that was closest to the side chain terminus, with ties broken by the number in the atom name (e.g. Cγ-Cδ1 instead of Cγ-Cδ2 for Leu). All negative order parameters (caused by high B-factors) were floored to 0. χ1 order parameters were floored for 7 residues, and side chain-end order parameters were floored for 23 residues. Per-residue 'apparent dynamic transition temperatures' were then calculated as the intersection between the <200 K and >200 K fit lines in order parameter versus temperature plots and floored to 0 K if necessary. The kernel density curve was fit with the *density* function in R (*Team, 2014*).

For extrapolation of fit lines in *Figure 6* , we used a fit to all data points or to just the high-temperature data points (≥240 K) for each residue, and extrapolated to the temperature at which order would be maximized (1 – $S^2$ = 0). To predict global behavior, at each temperature we averaged across all residues the predicted 1 – $S^2$ values from the fit, making sure to floor non-physical predicted values of 1 – $S^2$ < 0 to 0, as in (*Lee and Wand, 2001*).

## Acknowledgements

We thank Justin Biel, Bryn Fenwick, Robert Stroud, Ian Wilson, and Peter Wright for helpful conversations. DAK is supported by an A. P. Giannini Postdoctoral Research Fellowship. RAW is supported by a NSF Graduate Research Fellowship. MCT is supported by a BioXFEL Postdoctoral Fellowship. NKS and ASB are supported by NIH GM095887 and NIH GM102520. ATB and WIW acknowledge a Howard Hughes Medical Institute Collaborative Innovation Award (HCIA) that also provided funds

for the purchase of the microdiffractometer for the goniometer setup. HvdB is supported by the NIH Protein Structure Initiative U54GM094586 at the Joint Center for Structural Genomics and SLAC National Accelerator Laboratory grant SLAC-LDRD-0014-13-2. RET is supported by NSF MCB-1330685. JSF is a Searle Scholar, Pew Scholar, and Packard Fellow, and is supported by NIH OD009180, NIH GM110580, and NSF STC-1231306. Use of the Linac Coherent Light Source (LCLS), SLAC National Accelerator Laboratory, is supported by the U. S. Department of Energy, Office of Science, Office of Basic Energy Sciences under Contract No. DE-AC02-76SF00515. The use of the Cornell High Energy Synchrotron Source (CHESS) is supported by NSF DMR-1332208; the Macromolecular Diffraction at CHESS (MacCHESS) facility is supported by NIH GM103485; and the Stanford Synchrotron Radiation Lightsource, SLAC National Accelerator Laboratory, is supported by the U. S. Department of Energy, Office of Science, Office of Basic Energy Sciences under Contract No. DE-AC02-76SF00515. The SSRL Structural Molecular Biology Program is supported by the DOE Office of Biological and Environmental Research, and NIH GM103393.

## Additional information

### Competing interests

ATB: Reviewing editor, *eLife*.. The other authors declare that no competing interests exist.

### Funding

| Funder | Grant reference number | Author |
| --- | --- | --- |
| National Science Foundation | Graduate Research Fellowship | Rahel A Woldeyes |
| National Institutes of Health | GM095887 | Nicholas K Sauter |
| Kinship Foundation | Searle Scholar | James S Fraser |
| Pew Charitable Trusts | Pew Scholar | James S Fraser |
| David and Lucile Packard Foundation | Packard Fellow | James S Fraser |
| Howard Hughes Medical Institute | Collaborative Innovation Award (HCIA) | William I Weis Axel T Brunger |
| U.S. Department of Energy | DE-AC02-76SF00515 | Aina E Cohen |
| National Science Foundation | BioXFEL Postdoctoral Fellowship | Michael C Thompson |
| National Institutes of Health | GM102520 | Nicholas K Sauter |
| National Institutes of Health | GM094586 | Henry van den Bedem |
| National Science Foundation | MCB-1330685 | Robert E Thorne |
| SLAC National Laboratory | SLAC-LDRD-0014-13-2 | Henry van den Bedem |
| National Institutes of Health | OD009180 | James S Fraser |
| National Institutes of Health | GM110580 | James S Fraser |
| National Science Foundation | STC-1231306 | James S Fraser |
| National Science Foundation | DMR-1332208 | Robert E Thorne |
| National Institutes of Health | GM103485 | Robert E Thorne |
| National Institutes of Health | GM103393 | Aina E Cohen |
| A.P. Giannini Foundation | Postdoctoral Research Fellowship | Daniel A Keedy |

The funders had no role in study design, data collection and interpretation, or the decision to submit the work for publication.

### Author contributions

DAK, RET, Conception and design, Analysis and interpretation of data, Drafting or revising the article; LRK, MW, Conception and design, Acquisition of data, Drafting or revising the article; RAW, HvdB, JSF, Conception and design, Acquisition of data, Analysis and interpretation of data, Drafting or revising the article; JBH, ELB, SEM, JS, RA-M, SMS, HL, AG, AEC, Acquisition of data, Drafting or revising the article; MCT, JMH, NKS, Analysis and interpretation of data, Drafting or revising the article; ASB, AHVB, Acquisition of data, Analysis and interpretation of data, Drafting or revising the article; MU, WIW, Analysis and interpretation of data, Drafting or revising the article, Contributed unpublished essential data or reagents; ATB, Acquisition of data, Drafting or revising the article, Contributed unpublished essential data or reagents

### Author ORCIDs

Daniel A Keedy, http://orcid.org/0000-0002-9184-7586
Rahel A Woldeyes, http://orcid.org/0000-0003-0737-8383
William I Weis, http://orcid.org/0000-0002-5583-6150
Axel T Brunger, http://orcid.org/0000-0001-5121-2036
S Michael Soltis, http://orcid.org/0000-0003-4678-2995
Henry van den Bedem, http://orcid.org/0000-0003-2358-841X
James S Fraser, http://orcid.org/0000-0002-5080-2859

# Additional files

## Major datasets

The following datasets were generated:

| Author(s) | Year | Dataset title | Dataset URL | Database, license, and accessibility information |
|---|---|---|---|---|
| Keedy DA, Kenner LR, Warkentin M, Woldeyes RA, Thompson MC, Brewster AS, Van Benschoten AH, Baxter EL, Hopkins JB, Uervirojnang-koorn M, McPhillips SE, Song J, Mori RA, Holton JM, Weis WI, Brunger AT, Soltis M, Lemke H, Gonzalez A, Sauter NK, Cohen AE, van den Bedem H, Thorne RE, Fraser JS | 2015 | Multiconformer synchrotron model of CypA at 100 K | http://www.rcsb.org/pdb/explore/explore.do?structureId=4YUG | Publicly available at the RCSB Protein Data Bank (Accession no: 4yug). |
| Keedy DA, Kenner LR, Warkentin M, Woldeyes RA, Thompson MC, Brewster AS, Van Benschoten AH, Baxter EL, Hopkins JB, Uervirojnang-koorn M, McPhillips SE, Song J, Mori RA, Holton JM, Weis WI, Brunger AT, Soltis M, Lemke H, Gonzalez A, Sauter NK, Cohen AE, van den Bedem H, Thorne RE, Fraser JS | 2015 | Multiconformer synchrotron model of CypA at 150 K | http://www.rcsb.org/pdb/explore/explore.do?structureId=4YUH | Publicly available at the RCSB Protein Data Bank (Accession no: 4yuh). |

| | | | | |
|---|---|---|---|---|
| Keedy DA, Kenner LR, Warkentin M, Woldeyes RA, Thompson MC, Brewster AS, Van Benschoten AH, Baxter EL, Hopkins JB, Uervirojnang-koorn M, McPhillips SE, Song J, Mori RA, Holton JM, Weis WI, Brunger AT, Soltis M, Lemke H, Gon-zalez A, Sauter NK, Cohen AE, van den Bedem H, Thorne RE, Fraser JS | 2015 | Multiconformer synchrotron model of CypA at 180 K | http://www.rcsb.org/pdb/explore/explore.do?structureId=4YUI | Publicly available at the RCSB Protein Data Bank (Accession no: 4yui). |
| Keedy DA, Kenner LR, Warkentin M, Woldeyes RA, Thompson MC, Brewster AS, Van Benschoten AH, Baxter EL, Hopkins JB, Uervirojnang-koorn M, McPhillips SE, Song J, Mori RA, Holton JM, Weis WI, Brunger AT, Soltis M, Lemke H, Gon-zalez A, Sauter NK, Cohen AE, van den Bedem H, Thorne RE, Fraser JS | 2015 | Multiconformer synchrotron model of CypA at 240 K | http://www.rcsb.org/pdb/explore/explore.do?structureId=4YUJ | Publicly available at the RCSB Protein Data Bank (Accession no: 4yuj). |
| Keedy DA, Kenner LR, Warkentin M, Woldeyes RA, Thompson MC, Brewster AS, Van Benschoten AH, Baxter EL, Hopkins JB, Uervirojnang-koorn M, McPhillips SE, Song J, Mori RA, Holton JM, Weis WI, Brunger AT, Soltis M, Lemke H, Gon-zalez A, Sauter NK, Cohen AE, van den Bedem H, Thorne RE, Fraser JS | 2015 | Multiconformer synchrotron model of CypA at 260 K | http://www.rcsb.org/pdb/explore/explore.do?structureId=4YUK | Publicly available at the RCSB Protein Data Bank (Accession no: 4yuk). |
| Keedy DA, Kenner LR, Warkentin M, Woldeyes RA, Thompson MC, Brewster AS, Van Benschoten AH, Baxter EL, Hopkins JB, Uervirojnang-koorn M, McPhillips SE, Song J, Mori RA, Holton JM, Weis WI, Brunger AT, Soltis M, Lemke H, Gon-zalez A, Sauter NK, Cohen AE, van den Bedem H, Thorne RE, Fraser JS | 2015 | Multiconformer synchrotron model of CypA at 280 K | http://www.rcsb.org/pdb/explore/explore.do?structureId=4YUL | Publicly available at the RCSB Protein Data Bank (Accession no: 4yul). |

| | | | | |
|---|---|---|---|---|
| Keedy DA, Kenner LR, Warkentin M, Woldeyes RA, Thompson MC, Brewster AS, Van Benschoten AH, Baxter EL, Hopkins JB, Uervirojnang-koorn M, McPhillips SE, Song J, Mori RA, Holton JM, Weis WI, Brunger AT, Soltis M, Lemke H, Gonzalez A, Sauter NK, Cohen AE, van den Bedem H, Thorne RE, Fraser JS | 2015 | Multiconformer synchrotron model of CypA at 300 K | http://www.rcsb.org/pdb/explore/explore.do?structureId=4YUM | Publicly available at the RCSB Protein Data Bank (Accession no: 4yum). |
| Keedy DA, Kenner LR, Warkentin M, Woldeyes RA, Thompson MC, Brewster AS, Van Benschoten AH, Baxter EL, Hopkins JB, Uervirojnang-koorn M, McPhillips SE, Song J, Mori RA, Holton JM, Weis WI, Brunger AT, Soltis M, Lemke H, Gonzalez A, Sauter NK, Cohen AE, van den Bedem H, Thorne RE, Fraser JS | 2015 | Multiconformer synchrotron model of CypA at 310 K | http://www.rcsb.org/pdb/explore/explore.do?structureId=4YUN | Publicly available at the RCSB Protein Data Bank (Accession no: 4yun). |
| Keedy DA, Kenner LR, Warkentin M, Woldeyes RA, Thompson MC, Brewster AS, Van Benschoten AH, Baxter EL, Hopkins JB, Uervirojnang-koorn M, McPhillips SE, Song J, Mori RA, Holton JM, Weis WI, Brunger AT, Soltis M, Lemke H, Gonzalez A, Sauter NK, Cohen AE, van den Bedem H, Thorne RE, Fraser JS | 2015 | High-resolution multiconformer synchrotron model of CypA at 273 K | http://www.rcsb.org/pdb/explore/explore.do?structureId=4YUO | Publicly available at the RCSB Protein Data Bank (Accession no: 4yuo). |
| Keedy DA, Kenner LR, Warkentin M, Woldeyes RA, Thompson MC, Brewster AS, Van Benschoten AH, Baxter EL, Hopkins JB, Uervirojnang-koorn M, McPhillips SE, Song J, Mori RA, Holton JM, Weis WI, Brunger AT, Soltis M, Lemke H, Gonzalez A, Sauter NK, Cohen AE, van den Bedem H, Thorne RE, Fraser JS | 2015 | Multiconformer fixed-target X-ray free electron (XFEL) model of CypA at 273 K | http://www.rcsb.org/pdb/explore/explore.do?structureId=4YUP | Publicly available at the RCSB Protein Data Bank (Accession no: 4yup). |
| Fraser JS, Fraser J | 2015 | X-Ray Diffraction data from Cyclophilin A, source of 4YUO structure | https://data.sbgrid.org/dataset/68/ | Publicly available at the SRGrid Data Bank (Accession no: 10.15785/SBGRID/68). |

The following previously published dataset was used:

| Author(s) | Year | Dataset title | Dataset URL | Database, license, and accessibility information |
|---|---|---|---|---|
| Ke H | 1992 | Similarities and differences between human cyclophilin a and other beta-barrel structures. Structural refinement at 1.63 angstroms resolution | http://www.rcsb.org/pdb/explore/explore.do?structureId=2CPL | Publicly available at the RCSB Protein Data Bank (Accession no: 2CPL) |

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
