## [Decision Letter]

Thank you for sending your work entitled "Mapping the Conformational Landscape of a Dynamic Enzyme by Multitemperature and XFEL Crystallography" for consideration at *eLife*. Your article has been favorably evaluated by Michael Marletta (Senior Editor) and two reviewers, Stephen C Harrison and A Joshua Wand. The former is a member of our Board of Reviewing Editors.

This manuscript, a major technical achievement, gives an initial glimpse from a new perspective on several long-standing issues and controversies in protein biophysics. There are two significant observations – that local (mostly side chain) conformations are heterogeneous and in many cases show strong temperature dependence and that the temperature dependence is also heterogeneous.

Essential revisions:

A vast amount of work has gone into this report, and the reviewers agreed that it is a tour de force of experimental and computational effort. In its current form however, the manuscript makes it difficult to tease out where it leads. There seem to be two problems, both of which stem from efforts to fit the work into a framework that does not do it justice. Firstly, the authors interpret the results in terms of a transition at 200^o^ even when the data seem to show that there is no such sharp distinction. More importantly, the authors fail to draw obvious conclusions from the heterogeneity of the apparent temperature dependence and try to force it into an old model that seems to be disproven. Secondly, the inclusion of the XFEL experiments is unjustified.

Problem 1:

There are no data between 180° and 240°, the largest jump in the temperature range. This jump exaggerates the apparent break in the curves in Figure 5, and the relatively small (and imprecisely determined) occupancies of minor states at 180° below makes including those points in the "pseudo Van't Hoff" plots awkward. But within the large error bars that would obtain, many of the plots could probably include the lower temperature data and remain essentially linear (certainly true for my own hand replotting of Figure 5 as lnK vs. 1/T). In any case, it would seem that the heterogeneous response does not support the slaving model of Fraunfelder et al., since the natural thermal dependence of motion at individual sites will, as shown by Lee and Wand, give an average response typical of the scattering studies of old that started the dynamical transition and slaving models.

Problem 2:

The XFEL data are in many ways "scooped" by the 1.2 Å synchrotron data. In several figures, the XFEL points are lonely dots that simply fall on the expected curve. The one justification for inclusion of the XFEL data is the assertion that the dataset is damage free. But Figure 1—figure supplement 1 asserts that damage is minimal across data collection temperatures. There is some increase in R as exposure progresses, and the data were collected to avoid damage differences at different temperatures. Nonetheless, over the whole range of temperatures, the resolution is always better than that of the XFEL data, and had the synchrotron data been cut off at 1.75 Å spacing, it is possible that there would have been no evidence for any damage at any temperature.

Both these problems can most likely be addressed with some thorough reorganization and considerable rewriting of the manuscript, along the lines outlined below.

1) The Introduction should be about the real point of the paper (and the reason that it would be interesting to *eLife* readers): the internal dynamics of a protein. It should be addressed to someone interested in protein structure, but who is not acquainted with the earlier Frauenfelder and Petkso work, who has not read Lee and Wand (but needs to know about it), and who would like to understand whether the authors agree or disagree with the interpretations given by Kern to her studies of CypA dynamics. In its current form, the Introduction provides an unclear description of the inadequacies of current methods and the promise of XFEL data collection, and the paper then leaves a careful reader believing that the paper has in no way demonstrated that promise and if anything shown that current synchrotrons do better. Only around the fourth paragraph does the Introduction use the possibility of temperature jump (not used for any of the experiments here) as a reason to segue into what are (for the uninitiated) obscure references to Frauenfelder, Lee and Wand, and Eisenmesser et al.

2) In the presentation of Results, the same forced effort to justify the XFEL experiment shows through. For example, in the third paragraph, the failure of the XFEL data alone clearly to show the Leu98 alternative conformation gives rise to the argument "oh well, in the future, with less hardy crystals, we'll need it". We suggest at the end of this set of points how to include the XFEL data more modestly.

3) The Results are written for an "insider". What is "Ringer analysis", for example? If "Ringer" is the name of a program, it should be in all uppercase (like HKL2000, RELION, CHARMM, etc.). Surely there is a two or three word phrase that can summarize sampling the density for evidence of variation of torsion around chi1. Likewise, sentences like "our multiconformer models separate harmonic from non-harmonic contributions to flexibility" will be clear to MD savvy folks, but not to a sufficient fraction of the *eLife* readership to justify the journal as a good venue for a paper written this way.

4) As already mentioned, the interpretation of a sharp divide above and below 200° K seems forced. Sure, above 200° multiple conformations provide a better explanation than does a single one, but if two conformations are evident at 240, and the occupancy of the minor is not evident at 180°, that does not convince at least one reviewer that there is anything special about 200° or that there is something going on that is different from a smooth decrease in the minor state occupancy that simply falls below the detection threshold somewhere between 240° and 180°, but not in any way that suggests an abrupt transition (at least for the whole protein). This is the key point, both for presentation and analysis of the data and for the conclusions.

5) The paragraph about CypA in HIV is not particularly relevant, and the throw-away remark about an "arms race" should be rethought. Some evolutionary relationship between loop states and Gag binding might hold, but there are no lineage data cited even to hint at such a possibility.

6) A diagram is needed to explain what is meant by "tiered energy landscapes" – or at least what Frauenfelder meant by it. It is important to note that the tiers of Fraunfelder are related in timescale yet the heterogeneity of rotamer distributions seen here have been shown by many NMR studies to be on the same sub-nanosecond timescale.

7) To what extent are these data more or less consistent with the notions put forth by Lee and Wand, by Tilton et al, by Kern and coworkers, and ideally also with the (not mentioned) "sectors" analyzed by Ranganathan?

8) The reviewers are comfortable with inclusion of the XFEL data, but simply as another dataset. Some modest remark is of course justified, about how valuable the new postrefinement methods are for a good dataset, but Figure 8 should be removed.

Minor points:

Figure 6 fails to get across that the conformational heterogeneity and its temperature dependence are in direct opposition to the global "glass transition" popularized by the physics community. It is implied but not made explicit that this heterogeneous temperature dependence predicts the 200 K inflection using data obtained 100 degrees above it. In other words it is a direct consequence of the thermal behavior of internal motion of proteins in solution and has nothing really to do with a global concerted glass transition driven by the freezing of solvent.

The authors talk a lot about "dynamics", but this is not what they observe. Rather they observe presumably (near) equilibrium distributions. What is missing is a strong acknowledgement of timescale. For CypA, they somewhat casually link distributions to slower time scales (i.e. us-ms) rather than the ps-ns motions that actually govern side chain rotamer interconversion. The work of Fraser and Wright (Fenwick et al. 2014) addresses this point somewhat, but that work is only summarized in the present manuscript. Further to this point, the "hierarchical" model (beyond the tautological) is not necessarily consistent, since it requires higher "tiers" of motion to occur on increasingly slower timescales, which is something that this work cannot really address (although the NMR studies say it is not consistent, since the various classes of side chain motion observed occur on the same timescale).

The idea of frustration is overworked. In ubiquitin, one of only two proteins where an extensive temperature dependence of NMR-detected fast motion has been done, it has been shown by pressure perturbation that the coupling (effectively the source of frustration) between side chains is very limited.

---

## [Author Response]

Essential revisions:

A vast amount of work has gone into this report, and the reviewers agreed that it is a tour de force of experimental and computational effort. In its current form however, the manuscript makes it difficult to tease out where it leads. There seem to be two problems, both of which stem from efforts to fit the work into a framework that does not do it justice. Firstly, the authors interpret the results in terms of a transition at 200^o^ even when the data seem to show that there is no such sharp distinction. More importantly, the authors fail to draw obvious conclusions from the heterogeneity of the apparent temperature dependence and try to force it into an old model that seems to be disproven. Secondly, the inclusion of the XFEL experiments is unjustified.

We thank Professors Harrison and Wand for their thoughtful and considerate review. We have extensively reworked the Introduction and Discussion of the manuscript to focus on the divergent models of the relationship between the glass transition and functional dynamics. The results are now tighter and more linear in driving towards these points, reducing the emphasis of the XFEL dataset and accurately positioning it as an important positive control. Our detailed responses to these issues are contained below. Because of the extensive revisions to the manuscript we only include outlines of the revisions.

*Problem 1:*

There are no data between 180° and 240°, the largest jump in the temperature range.

Technical issues made it difficult to collect data in this temperature range.

This jump exaggerates the apparent break in the curves in Figure 5, and the relatively small (and imprecisely determined) occupancies of minor states at 180° below makes including those points in the "pseudo Van't Hoff" plots awkward. But within the large error bars that would obtain, many of the plots could probably include the lower temperature data and remain essentially linear (certainly true for my own hand replotting of Figure 5 as lnK vs. 1/T). In any case, it would seem that the heterogeneous response does not support the slaving model of Fraunfelder et al., since the natural thermal dependence of motion at individual sites will, as shown by Lee and Wand, give an average response typical of the scattering studies of old that started the dynamical transition and slaving models.

We have deleted the Van’t Hoff plots because we agree that the precision of the occupancy estimate is problematic and can potentially be obscured with increasing B-factors. This uncertainty is a major advantage of the crystallographic order parameter approach, which allows the interaction between the B-factor and occupancy estimates to be accounted in one metric.

In any case, it would seem that the heterogeneous response does not on the face of it support the slaving model of Fraunfelder et al since the natural thermal dependence of motion at individual sites will, as shown by Lee and Wand, give an average response typical of the scattering studies of old that started the dynamical transition and slaving models.

We will argue below for an interpretation that reconciles many aspects of the two models. The crystal solvent does indeed undergo some transition (or more likely multiple transitions) that arrests the progression of disorder in the protein sidechains. The heterogeneity in the temperature this occurs at is reflected in our original analysis of the intersection temperatures. In addition, our data confirm the central ideal of Lee and Wand: that a transition can be observed by monitoring the average disorder after extrapolating the displacements across all protein sites (as in Figure 4 of Lee and Wand, 2001, with individual order parameters bounded between 0 and 1). In our data, however, a flattening transition of the average behavior is only seen when the high temperature data are used (red line) and not when all data are used (purple line). This transition occurs at a lower temperature than in Lee and Wand. In contrast, plotting the average behavior of our observed data (data points) also reveals a flattening transition, but with a higher degree of ordering (and between the 180-240 K data points) than the extrapolation of what is projected from Lee and Wand.

We interpret the origin of this flattening as the solvent glassy state preventing the evolution of further motion in the crystal. The barrier for the protein to transition between different conformations is therefore raised above the projection we would expect based on the simple temperature extrapolation (red line). However, we note that many distinct transitions likely occur across this temperature range as different parts of the bulk solvent and different ordered solvent regions become much more strongly ordered.

These observations thus reconcile the two models. The bulk solvent likely does undergo a complex transition that arrests protein motions, and the average protein sidechain likely does undergo a transition that is driven by thermal depopulation of alternative conformations and unrelated to a global bulk solvent transition. It is also important to note that the bulk solvent transition is not global; even Teeter, Yamano, Stec, and Mohanty, PNAS, 2001, indicate distinct local transition temperatures for waters based on X-ray data. In summary, the discordance in the observed vs. projected average order strongly suggests that the sidechain motions reaching maximum order and the sidechain motions arresting arise from different underlying causes.

A plot has now been added as a panel of Figure 6 (Figure 6), which now focuses on the non-global nature of the averaged transition measured by this comparison and the intersection analysis presented in the previous version. This topic also forms a major part of the revised Discussion.

*Problem 2:*

The XFEL data are in many ways "scooped" by the 1.2 Å synchrotron data. In several figures, the XFEL points are lonely dots that simply fall on the expected curve. The one justification for inclusion of the XFEL data is the assertion that the dataset is damage free. But Figure 1—figure supplement 1 asserts that damage is minimal across data collection temperatures. There is some increase in R as exposure progresses, and the data were collected to avoid damage differences at different temperatures. Nonetheless, over the whole range of temperatures, the resolution is always better than that of the XFEL data, and had the synchrotron data been cut off at 1.75 Å spacing, it is possible that there would have been no evidence for any damage at any temperature.

We think this positive control is important because we have experienced skepticism about the influence of radiation damage on the functionally important alternative conformations we observe only at higher temperatures. In the revised manuscript, we have emphasized the control aspect of the XFEL work and de-emphasized its role in general.

*Both these problems can most likely be addressed with some thorough reorganization and considerable rewriting of the manuscript, along the lines outlined below. 1) The Introduction should be about the real point of the paper (and the reason that it would be interesting to* eLife *readers): the internal dynamics of a protein. It should be addressed to someone interested in protein structure, but who is not acquainted with the earlier Frauenfelder and Petkso work, who has not read Lee and Wand (but needs to know about it), and who would like to understand whether the authors agree or disagree with the interpretations given by Kern to her studies of CypA dynamics. In its current form, the Introduction provides an unclear wishy-description of the inadequacies of current methods and the promise of of XFEL data collection, and the paper then leaves a careful reader believing that the paper has in no way demonstrated that promise and if anything shown that current synchrotrons do better. Only around the fourth paragraph does the Introduction use the possibility of temperature jump (not used for any of the experiments here) as a reason to segue into what are (for the uninitiated) obscure references to Frauenfelder, Lee and Wand, and Eisenmesser et al.*

We have re-written the Introduction extensively. The major themes of the paragraphs in this shortened introduction are:

a) General problems in protein dynamics and links to function;

b) Problems with cryocrystallography;

c) The two classic models of the glass transition and the recent ssNMR results;

d) How Ringer, multiconformer X-ray models, and crystallographic order parameters overcome the limitations of previous X-ray studies based on B-factors;

e) How the functional dynamics of CypA provide a new model system for testing the links between functional dynamics and the glass transition.

2) In the presentation of Results, the same forced effort to justify the XFEL experiment shows through. For example, in the third paragraph, the failure of the XFEL data alone clearly to show the Leu98 alternative conformation gives rise to the argument "oh well, in the future, with less hardy crystals, we'll need it". We suggest at the end of this set of points how to include the XFEL data more modestly.

We have simplified the presentation of the XFEL experiments, properly positioning the data as a positive control.

*3) The Results are written for an "insider". What is "Ringer analysis", for example? If "Ringer" is the name of a program, it should be in all uppercase (like HKL2000, RELION, CHARMM, etc.). Surely there is a two or three word phrase that can summarize sampling the density for evidence of variation of torsion around chi1. Likewise, sentences like "our multiconformer models separate harmonic from non-harmonic contributions to flexibility" will be clear to MD savvy folks, but not to a sufficient fraction of the* eLife *readership to justify the journal as a good venue for a paper written this way.*

We have included simpler explanations of the Ringer and multiconformer modeling procedures and included a more simple definition of harmonic and non-harmonic contributions. In addition, we made other textual changes to increase the clarity of the manuscript. The uppercase naming convention derives from acronyms (e.g. REgularised LIkelihood OptimisatioN). Since Ringer doesn’t stand for anything other than the name Ringer, we have not included it in uppercase.

*4) As already mentioned, the interpretation of a sharp divide above and below 200° K seems forced. Sure, above 200° multiple conformations provide a better explanation than does a single one, but if two conformations are evident at 240, and the occupancy of the minor is not evident at 180°, that does not convince at least one reviewer that there is anything special about 200° or that there is something going on that is different from a smooth decrease in the minor state occupancy that simply falls below the detection threshold somewhere between 240° and 180°, but not in any way that suggests an abrupt transition (at least for the whole protein). This is the key point, both for presentation and analysis of the data and for the conclusions.*

Indeed, our data do not favor a single transition temperature (that is what is shown in Figure 6). However, our data do suggest that the temperature dependent ordering is strongly restricted at 180 K and below. The slopes are significantly different on either side of this (relatively broad) transition range (the average slope is 2.5x10^-4^ (1-S^2^)/K for 180K and below, and 2.6x10^-3^ (1-S^2^)/K for 240K and above; p=1x10^-62^ by a paired T-test).

5) The paragraph about CypA in HIV is not particularly relevant, and the throw-away remark about an "arms race" should be rethought. Some evolutionary relationship between loop states and Gag binding might hold, but there are no lineage data cited even to hint at such a possibility.

We have deleted this section. For the reviewers’ interest, we point to the excellent papers by Leo James, cited in the original manuscript, on the relationship between loop states, CypA/TRIMCypA lineage, and binding.

*6) A diagram is needed to explain what is meant by "tiered energy landscapes" – or at least what Frauenfelder meant by it. It is important to note that the tiers of Fraunfelder are related in timescale yet the heterogeneity of rotamer distributions seen here have been shown by many NMR studies to be on the same sub-nanosecond timescale.*

We explain more about the population shuffling model and tiers below. Briefly, the exchange of individual sidechain conformations may be fast on the ns timescale, but the dominant conformational exchange that shuffles these conformations is on the ms timescale. We suggest below that the rotameric switch of Phe113 is a dominant contributor to the NMR dynamics.

*7) To what extent are these data more or less consistent with the notions put forth by Lee and Wand, by Tilton et al, by Kern and coworkers, and ideally also with the (not mentioned) "sectors" analyzed by Ranganathan?*

The discussion is now focused on the following points:

a) A general summary of our major findings on CypA;

b) New insights into the relationship between energy landscapes, the glass transition, and protein function;

c) Glass transition models;

d) Relationship of CypA motion to catalysis and the population shuffling models (Kern and our prior work with her group);

e) The longer term potential for multitemperature and XFEL data to reveal coupled motions and evolutionary selection in proteins (Ranganathan and others).

8) The reviewers are comfortable with inclusion of the XFEL data, but simply as another dataset. Some modest remark is of course justified, about how valuable the new postrefinement methods are for a good dataset, but Figure 8 should be removed.

We have deleted the figure and refocused the discussion of the XFEL data.

*Minor points:Figure 6 fails to get across that the conformational heterogeneity and its temperature dependence are in direct opposition to the global "glass transition" popularized by the physics community. It is implied but not made explicit that this heterogeneous temperature dependence predicts the 200 K inflection using data obtained 100 degrees above it. In other words it is a direct consequence of the thermal behavior of internal motion of proteins in solution and has nothing really to do with a global concerted glass transition driven by the freezing of solvent.*

We have replaced the original Figure 6. Figure 6 now plots disorder vs. temperature for one representative residue. Figure 6 now plots the distribution of the intersection temperature between the two lines across the protein. Figure 6 now plots the new analysis bridging different models of the dynamical transition. To address the reviewer’s criticism, we have moved the original “small multiples” Figure 6 to figure supplements, to support the new Figure 6.

The authors talk a lot about "dynamics", but this is not what they observe. Rather they observe presumably (near) equilibrium distributions.

The reviewers are correct that the observations we make are of “conformational heterogeneity” near equilibrium. We are careful not refer to dynamics at all in the Discussion of the X-ray results directly.

What is missing is a strong acknowledgement of timescale. For CypA, they somewhat casually link distributions to slower time scales (i.e. us-ms) rather than the ps-ns motions that actually govern side chain rotamer interconversion. The work of Fraser and Wright (Fenwick et al. 2014) addresses this point somewhat, but that work is only summarized in the present manuscript. Further to this point, the "hierarchical" model (beyond the tautological) is not necessarily consistent, since it requires higher "tiers" of motion to occur on increasingly slower timescales, which is something that this work cannot really address (although the NMR studies say it is not consistent, since the various classes of side chain motion observed occur on the same timescale).

Indeed, our earlier analysis (Fraser et al, 2009), performed in collaboration with the Kern lab, focused on the correspondence between the two states observed by NMR relaxation studies (with an exchange rate of ~1000s-1) and the two dominant alternative conformations observed from room-temperature X-ray studies, which obviously do not specify the timescale. The heterogeneity in the temperature response indicates that the sidechain conformational heterogeneity is not as strongly correlated as we had previously suspected. In the revised manuscript, we propose that many CypA sidechains are in relatively fast exchange, but that a slower process (most probably the coupled sidechain and backbone “backrub” motion of Phe113) shuffles these conformations. This phenomenon of “population shuffling” has been suggested based on ubiquitin (Smith et al, 2015) and is also consistent with the observation that simplistic model of clashes in CypA cannot classify two distinct states (van den Bedem et al, 2013).

The idea of frustration is overworked. In ubiquitin, one of only two proteins where an extensive temperature dependence of NMR-detected fast motion has been done, it has been shown by pressure perturbation that the coupling (effectively the source of frustration) between side chains is very limited.

We have removed the original sentence that contains the reference to frustration, and now mention it only with respect to glassy systems in general.